# Label-free identification of protein aggregates using deep learning

Khalid A. Ibrahim [1,2], Kristin S. Grußmayer [3] ✉, Nathan Riguet [2], Lely Feletti [1], Hilal A. Lashuel [2] ✉ & Aleksandra Radenovic [1] ✉

Protein misfolding and aggregation play central roles in the pathogenesis of various neurodegenerative diseases (NDDs), including Huntington's disease, which is caused by a genetic mutation in exon 1 of the Huntingtin protein (Httex1). The fluorescent labels commonly used to visualize and monitor the dynamics of protein expression have been shown to alter the biophysical properties of proteins and the final ultrastructure, composition, and toxic properties of the formed aggregates. To overcome this limitation, we present a method for label-free identification of NDD-associated aggregates (LINA). Our approach utilizes deep learning to detect unlabeled and unaltered Httex1 aggregates in living cells from transmitted-light images, without the need for fluorescent labeling. Our models are robust across imaging conditions and on aggregates formed by different constructs of Httex1. LINA enables the dynamic identification of label-free aggregates and measurement of their dry mass and area changes during their growth process, offering high speed, specificity, and simplicity to analyze protein aggregation dynamics and obtain high-fidelity information.

Neurodegenerative diseases (NDDs), such as Alzheimer's disease (AD), Parkinson's disease (PD), and Huntington's disease (HD), are major global health concerns afflicting tens of millions of patients worldwide and are currently incurable[1,2]. They are characterized by progressive damage to the structure and function of neurons and the accumulation of misfolded proteins in specific brain regions and neuronal populations[3–6]. Although these misfolded protein aggregates represent the hallmark of several NDDs, including AD, PD, and HD, our understanding of their mechanisms of formation and role in the disease pathogenesis remains incomplete. While some studies suggest that their formation is a major driver of neurodegeneration, other studies propose that they represent neuroprotective mechanisms that lead to inactivation of toxic aggregates[7–12]. Furthermore, the nature of toxic species remains unknown. One primary reason for this knowledge gap is the lack of tools and methods that allow for direct monitoring of the different phases of protein misfolding and aggregation, leading to the formation of the pathological hallmarks found in the brains of affected individuals.

To enable monitoring protein aggregation in living cells or animals, fluorescent proteins (FPs), such as GFP, YFP, mCherry, and others, are usually fused to the C- or N-terminus of the protein of interest. Protein aggregation is usually seen as the transition from diffuse signal to the formation of puncta or foci structures. However, as useful and valuable as FPs can be, they also have limitations. Several studies have shown that fluorescently tagged proteins exhibit altered biochemical, biophysical, or cellular properties[13–22]. In the context of protein aggregation linked to NDDs, the addition of fluorescent proteins to proteins of different sizes has been shown to alter not only the kinetics of aggregation and the final size of the aggregates, but also the interactome of the aggregate and their ultrastructural organization in the final inclusions that accumulate in cells or brain tissues. This discrepancy has been observed in several NDD-associated proteins,

[1]Laboratory of Nanoscale Biology, École Polytechnique Fédérale de Lausanne (EPFL), Lausanne, Switzerland. [2]Laboratory of Molecular and Chemical Biology of Neurodegeneration, École Polytechnique Fédérale de Lausanne (EPFL), Lausanne, Switzerland. [3]Department of Bionanoscience and Kavli Institute of Nanoscience Delft, Delft University of Technology, Delft, Netherlands. ✉e-mail: k.s.grussmayer@tudelft.nl; hilal.lashuel@epfl.ch; aleksandra.radenovic@epfl.ch

including alpha-synuclein[18,19], tau[20], amyloid beta[21,22] (Aβ), and exon1 of the Huntingtin (Htt) protein[14–17] (Httex1), one of the primary components of intracellular protein aggregates found in Huntington's disease post-mortem brains[23–25]. Httex1 overexpression in multiple cellular and animal models of HD recapitulates many of the key features of HD human pathology, including Htt aggregation and inclusion formation, neurodegeneration, and brain atrophy[8,26–30].

In a recent study[14], we compared the biochemical and ultrastructural properties of cellular inclusions formed by mutant Httex1, with expanded polyglutamine (polyQ) repeats of 16-72Q, in the absence or presence of GFP on the C-terminus of the protein. Our results showed that the addition of GFP to mutant Httex1 not only altered the biochemical and ultrastructural properties of the inclusions but also their toxic properties and interactions with cellular organelles. For example, the inclusions formed by unlabeled mutant Httex1 displayed distinctive core and shell structures that contain lipids and membranous organelles and within which the Htt fibrils colocalize with and interact with the endoplasmic reticulum (ER) structure and membranous structures. In contrast, mutant Httex1-GFP formed a mesh of cytoplasmic fibrils that exhibited minimal interactions with cellular organelles (Fig. 1a). These observations are consistent with previous studies showing that fusion of large proteins to mutant Httex1, such as GFP, alter their dimensions as well as their mechanical and surface properties. GFP-labeled fibrils were ~3 nm thicker[15] and exhibited increased stiffness[16] compared to the unlabeled Httex1 fibrils. Using coarse-grained simulations, we recently showed that fluorescent tags like GFP cause a size-dependent surface occlusion and thus alter the fibril's interactome[17]. These observations could explain the distinct interactome observed for untagged and GFP-tagged mutant Htt inclusions in mammalian cells and primary neurons[14]. While it is tempting to attribute these changes in the biophysical and biochemical properties to the large size of fluorescent proteins, it has been shown that even the fusion of small peptide tags like HA to different N-terminal Htt fragments could markedly change their aggregation and toxic properties[31]. Altogether, these observations underscore the critical importance of developing label-free methods that enable investigation of the mechanisms of protein aggregation and inclusion formation of the native proteins.

Although recent studies have suggested using label-free methods, such as autofluorescence and Raman microscopy[32–35], to monitor protein aggregation in cells, these methods lack the specificity and contrast needed to analyze them sufficiently, often lack the needed temporal resolution, or require deuterium labeling. Quantitative phase imaging (QPI)[36–38] is an attractive label-free microscopy modality that produces a quantitative image of an unlabeled biological specimen based on the phase shift at each pixel in the field of view (FOV), enabling the extraction of quantitative parameters, such as the dry mass or morphology. Since low-intensity illumination can be used, quantitative information can be obtained from live specimens with minimal photodamage and photobleaching (compared to fluorescence imaging) at rapid time scales. However, the lack of a specific label complicates distinguishing between different structures inside cells. In recent years, the concept of virtual staining has emerged[39–42], in which a deep learning model is taught to map between a label-free imaging modality (e.g., brightfield, QPI, or auto-fluorescence) and fluorescence imaging, such that the model can predict the fluorescence signal strictly from the label-free input, yielding the necessary contrast and specificity. A recent technique called VISTA uses sample expansion to achieve super-resolved Raman imaging and virtual labeling of Aβ plaques and mutant Htt aggregates[43]. However, this technique was not validated on label-free Htt aggregates, was not quantitatively validated on any Htt aggregates, had a large variation in Aβ-plaque model performance (roughly between 30% and 90%), requires substantial sample manipulation and perturbation during expansion, and is fundamentally incompatible with live-cell imaging.

In this work, we employ the concept of virtual labeling[39–42] on a well-characterized cellular model of mutant Httex1 aggregation and inclusion formation by training a neural network on data collected using widely available imaging modalities (brightfield/QPI and widefield fluorescence) to generate label-free identification of NDD-associated aggregates (LINA) models (Fig. 1b). We quantitatively validate the LINA models and demonstrate that they can be used to accurately identify unlabeled mutant Httex1 aggregates with areas as small as $3\,\mu m^2$ and at exposure times as low as 3 ms. By applying our neural network models to identify aggregates formed by different label-free mutant Httex1 constructs, we are able to measure and compare their dry masses. We can also identify aggregates from live-cell imaging data and are able to measure the dry mass and area of aggregates as they form. We experimentally validate the models' robustness to different acquisition settings and cell lines, showing the high applicability of the method. The aforementioned issues related to labeling illustrate the need for label-free techniques for identifying and

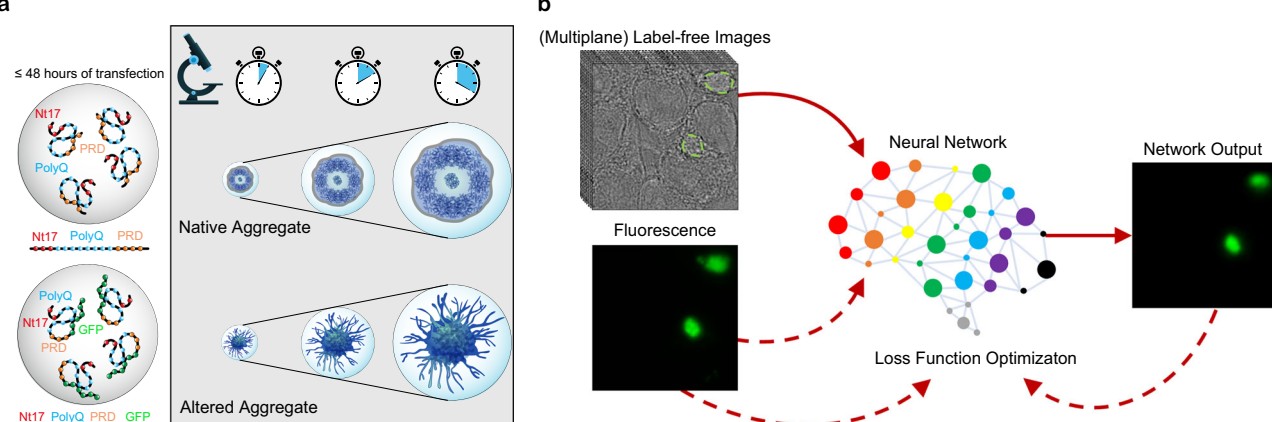

**Fig. 1 | Label-free identification of NDD-associated aggregates (LINA). a** The NDD-associated Httex1 protein forms aggregates in cells within 48 h. When the protein is unlabeled, these aggregates have a core and shell ultrastructure. Labeled Httex1 (e.g., with GFP) forms altered aggregates that lack this structure, instead resembling a mesh of fibrils, and have altered biochemical and biophysical properties (e.g., different proteome composition, stiffness, and fibril length). **b** To enable label-free imaging of unaltered Httex1 aggregates, we trained a neural network to map between label-free transmitted-light (brightfield or quantitative phase) image inputs (single or multiple planes) and fluorescence images, such that the network is then able to identify aggregates using only the label-free input. Dashed arrows represent training-only steps.

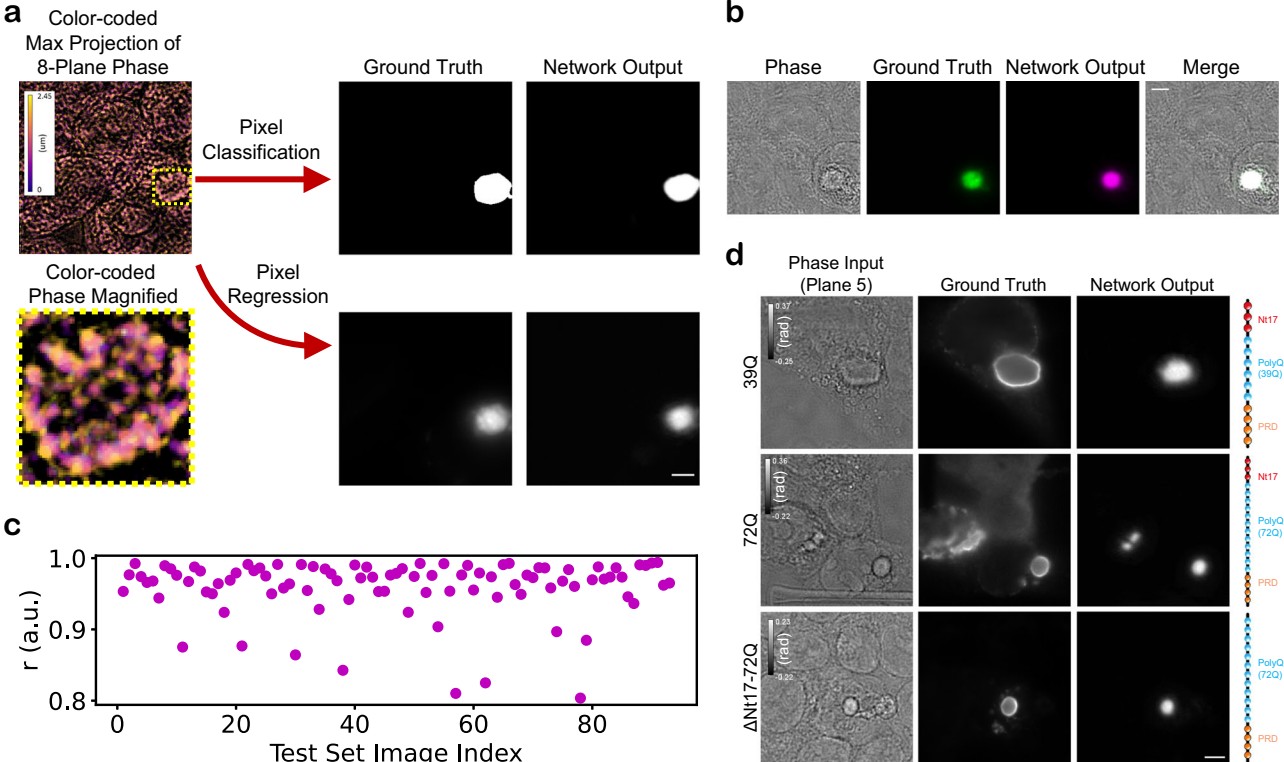

**Fig. 2 | Validation of deep learning models for label-free identification of Httex1 protein aggregates. a** Convolutional neural networks for both pixel classification and regression have been trained using pixel-registered pairs of eight-plane quantitative phase images and maximum-projected eight-plane fluorescence images or corresponding segmented masks. The eight-plane input images are represented by a color-coded maximum $z$-projection image; the aggregate in the image is highlighted by the dashed square and is shown magnified. The phase signal was thresholded ($T = 0$ rad) prior to color-coding to enhance the contrast. The labels used are generated as the maximum $z$-projection of the fluorescence images in the eight planes for the pixel regression network and the corresponding segmented masks (Otsu thresholding) for pixel classification. **b** Test set prediction example, with images of the phase, fluorescence, network output, and a merge of all three, showing where the three images colocalize in white. This example is representative of the other examples in the test set ($n = 105$ acquisitions from three independent experiments). **c** Quantitative validation of the pixel regression model using the Pearson correlation coefficient ($r$), showing a high correlation with the ground truth (-0.96 mean). **d** The regression network was further validated on label-free Httex1 aggregates. Despite being trained only on a 72Q-GFP construct of Httex1, the model works well on label-free constructs of different polyQ repeat lengths (39Q, 72Q) and types (72Q with a truncated Nt17 domain). This validation was repeated over three independent experiments. Scale bars: 5 μm.

analyzing native protein aggregates to avoid tag-induced alterations. The label-free method we have developed to visualize and analyze protein aggregates demonstrates, for the first time to our knowledge, the applicability of virtual labeling on unlabeled and unaltered protein aggregates in living cells, with rapid speed, high accuracy, and relative simplicity. LINA paves the way for more accurate analysis of NDD-associated protein aggregates, better recapitulating their true neurobiological nature and offering higher-fidelity information.

## Results
### Label-free identification of aggregates formed by different Httex1 constructs with high accuracy
The choice of a well-controlled biological model system is crucial for the development of robust and reproducible image analysis tools. To apply virtual labeling on NDD-associated aggregates and develop LINA models, we chose to use a cellular model of HD that is well-characterized. As reported previously, when a mutant Httex1 construct with a polyQ repeat length ≥39 is overexpressed in HEK 293 (HEK) cells, aggregates are formed within a period of 48 h. With our custom-built, multi-modal, multi-plane microscope[44], it is possible to acquire ultrafast 4D brightfield and fluorescence images using an image-splitting prism that introduces path length differences in the detection path. This prism allows the acquisition of an image stack consisting of eight $z$-planes simultaneously. An algorithm is then used to retrieve phase information from the brightfield image stack using

Fourier filtering, transforming it into a QPI stack. We used this microscope to collect a dataset of over 1000 eight-plane brightfield and fluorescence pairs of images of fixed HEK cells overexpressing a mutant Httex1 construct with a polyQ repeat length of 72, fused to GFP (Httex1-72Q-GFP).

We processed the dataset to obtain pixel-registered eight-plane QPI (which we also refer to as 'phase' images) and fluorescence images. We then used a subset of the dataset (the training and validation sets) to train a convolutional neural network (CNN) (with a U-Net architecture[45], more information is provided in the "Methods" section) and produce LINA models for both pixel regression and pixel classification (Fig. 2a). Here, each ground truth label is generated as the maximum $z$-projection of each eight-plane image stack so that it incorporates information from the eight planes, reducing the complexity of the mapping. We aim to achieve 2D, rather than 3D, identification of the aggregates, for several reasons. A major reason is that this lowers the memory constraints considerably and enhances the accessibility of the method. 3D prediction is a challenging task, as described in similar prior work[39]. Using our 2D-identification models, an aggregate can be identified in 2D space first, and then its position within the other planes can be inferred from the QPI images afterward. The pixel classification model is trained on segmentation masks generated from the maximum $z$-projection images, and it classifies the pixels into either 1's or 0's, i.e., part of an aggregate or not. The pixel regression model is more general in that it is trained directly on the

maximum *z*-projection fluorescence images, and can predict intensity values, rather than just 1's or 0's. In Fig. 2a, the phase image is a color-coded maximum *z*-projection, where each z-plane is represented using a different color, illustrating the spread of information available in the eight planes. The network outputs were observed to match the ground truths quite well, as can be verified further in Fig. 2b. The phase images are represented in this figure and other figures by one of the central planes, i.e., plane 4 or 5. Further visual test-set examples for both model types are shown in Supplementary Figs. 1 and 2.

To precisely quantify how well the LINA pixel-regression model performs, we measured the Pearson correlation coefficient (*r*) between the network output and the ground truth (Fig. 2c). The correlation coefficient is consistently high (0.959 mean) over the entire test set, a set of images which the model had never seen before, illustrating the high reliability of the model. We further evaluated the quantitative performance of the model by measuring *r* and the normalized mean squared error (NMSE), the quantity used for training (unnormalized), computed only within the regions where there are aggregates (Supplementary Fig. 3a, b), thereby more precisely measuring the degree of true positives/false negatives (Methods). The correlation remains very high (0.955 mean), and the NMSE also consistently shows relatively low errors (0.106 mean). The pixel classification model also performs well, achieving an F1-score of 0.9 and a mean Jaccard index of 0.78. The pixel-regression model is slightly more accurate when segmentations are produced from its regression predictions (using Otsu thresholding), with a mean Jaccard index of 0.81 and a mean Dice loss of 0.89 (Supplementary Fig. 3c, d). Therefore, we focus on the regression model for our following experiments and quantifications. We used the Pearson correlation coefficient to compare the total intensity within the aggregates in the prediction images versus the ground truth, which was computed as 0.91, suggesting a very high correlation in this regard as well. Supplementary Fig. 4 shows LINA's performance on two negative controls (cells expressing 16Q-GFP and eGFP, respectively), where aggregates are not produced. LINA successfully predicts the absence of aggregates in both cases.

To determine if the models work on the unlabeled protein aggregates and if they generalize to other constructs than the one it was trained on, we sought to validate the network on different label-free constructs of mutant Httex1. The constructs we tested are unlabeled mutant Httex1 with a polyQ repeat length of 39 (Httex1-39Q), unlabeled mutant Httex1 with a polyQ repeat length of 72 (Httex1-72Q), and unlabeled mutant Httex1 with a polyQ repeat length of 72 with a truncated Nt17 domain (Httex1-ΔNt17-72Q). These constructs, which either have a shorter polyQ repeat length or have a truncated Nt17 domain, have been shown to influence the kinetics of aggregation, as well as the ultrastructural properties of Httex1 inclusions when comparing 39Q and 72Q[14]. Therefore, it was important to verify if LINA can identify aggregates formed using these constructs despite the differences in aggregate properties. To obtain ground truth, we performed immunocytochemistry (ICC) on the three constructs and labeled the aggregates with an anti-Htt antibody raised against the Proline-rich domain (PRD) (MAB5492, Millipore). We acquired QPI images of the same cells before and after ICC, using grid-marked coverslips, to ensure that LINA works well in both cases. Figure 2d shows the input, ground truth and network output for each construct. For all three constructs, the network correctly identifies the aggregates, whose peripheries are shown in the ground truth images; this is because the antibody we used is known to show a strong immunoreactivity to the periphery of the aggregate rather than the core[14], while the network was trained solely on GFP-labeled protein aggregates. Although the CNN was only trained on images of Httex1-72Q-GFP, the model is nonetheless generalizable to different kinds of label-free Httex1 constructs. For Httex1-ΔNt17-72Q, the model is able to discern between rather small closely neighboring aggregates (areas of around 3 μm²), instead of seeing them as one larger aggregate.

## Robustness and generalizability of LINA

Testing the robustness of LINA in various image acquisition, input types, label quality, and cell line configurations is vital to ensure the models' robustness and generalizability to different kinds of data and to facilitate the usability of the models by new users. We observed, in test-set examples that are being seen by the network for the first time, that the pixel-classification model can identify aggregates that are missing in the label due to segmentation errors (Supplementary Fig. 5a). The model outputs outperform direct segmentation from the ground truth, showing that it can learn very well what constitutes an aggregate despite some erroneous labels.

The signal-to-noise ratio (SNR) of microscopy images is a key determinant of how well image analysis pipelines and models perform. Increasing amounts of noise can reduce the accuracy and precision of segmentation and classification models, leading to incorrect or inconsistent results. All our models were trained on images taken at a 50 ms exposure time and a fixed illumination, so we wanted to test whether it is possible to identify aggregates from images taken at lower SNRs. We verified LINA's robustness to noise both using simulated noise and experimentally (Supplementary Fig. 5b–h). Through a simple pre-filtering step, we show that it is possible to identify aggregates even at excessive amounts of noise (variance of 0.48 rad²) where nothing can be visually identified in the image anymore. By adding noise specifically to the aggregate regions and with the same pre-filtering step, we show that LINA has a high potential to be applied to aggregates of varying optical properties. We observe, both visually and quantitatively, that the performance degrades when noise is added that is larger than or equal to 0.5 rad², so we consider this the acceptable range for our model, though it can be extended in the future through transfer learning. We then experimentally verified the model's robustness to different SNRs, as we measured a consistently high SSIM at various exposure times, as low as 3 ms, without the need for pre-filtering, despite the model being solely trained on images acquired at a 50 ms exposure time.

Another aspect we wanted to assess was the models' generalizability to different cell lines than what we used for training (HEK). Therefore, we quantified LINA's ability to identify Httex1-72Q-GFP aggregates expressed in HeLa cells, and the model consistently produced highly accurate predictions, as quantified by measuring the Pearson correlation coefficient and the NMSE between the network outputs and the ground truth images (Fig. 3a, b). Figure 3c, d show example images with the highest and lowest correlation coefficients, respectively. The lowest-correlation example still shows excellent performance, as both aggregates can be identified by the model. The reason for the slightly lower correlation could possibly be that having two aggregates in one FOV complicates the prediction of the true intensity levels for both aggregates at the same time, as we observed that the example with the highest NMSE (worst performance) also had two aggregates in the FOV, though this was also accurately predicted. These results indicate the high potential for LINA to be applied in other cell lines, greatly enhancing its generalizability.

To test the potential of generalizing LINA to various kinds of protein aggregates involved in other NDDs, which could vary in terms of circularity and heterogeneity, we simulated a non-circular, chimeric test set and tested LINA's performance on it. This was done by replacing parts of aggregates in the test set with images of cells/background (for both the input and label), and then testing the ability of the pixel-regression model to correctly identify the heterogeneous aggregates (Supplementary Fig. 6). The Pearson correlation coefficient (mean 0.89) and NMSE (mean 0.08) show excellent performance even with these perturbations, demonstrating the high potential of generalizing LINA to different kinds of protein aggregates, of varying morphologies and structure.

We also evaluated LINA's dependence on the number of planes of the input by training new pixel-regression models using QPI inputs

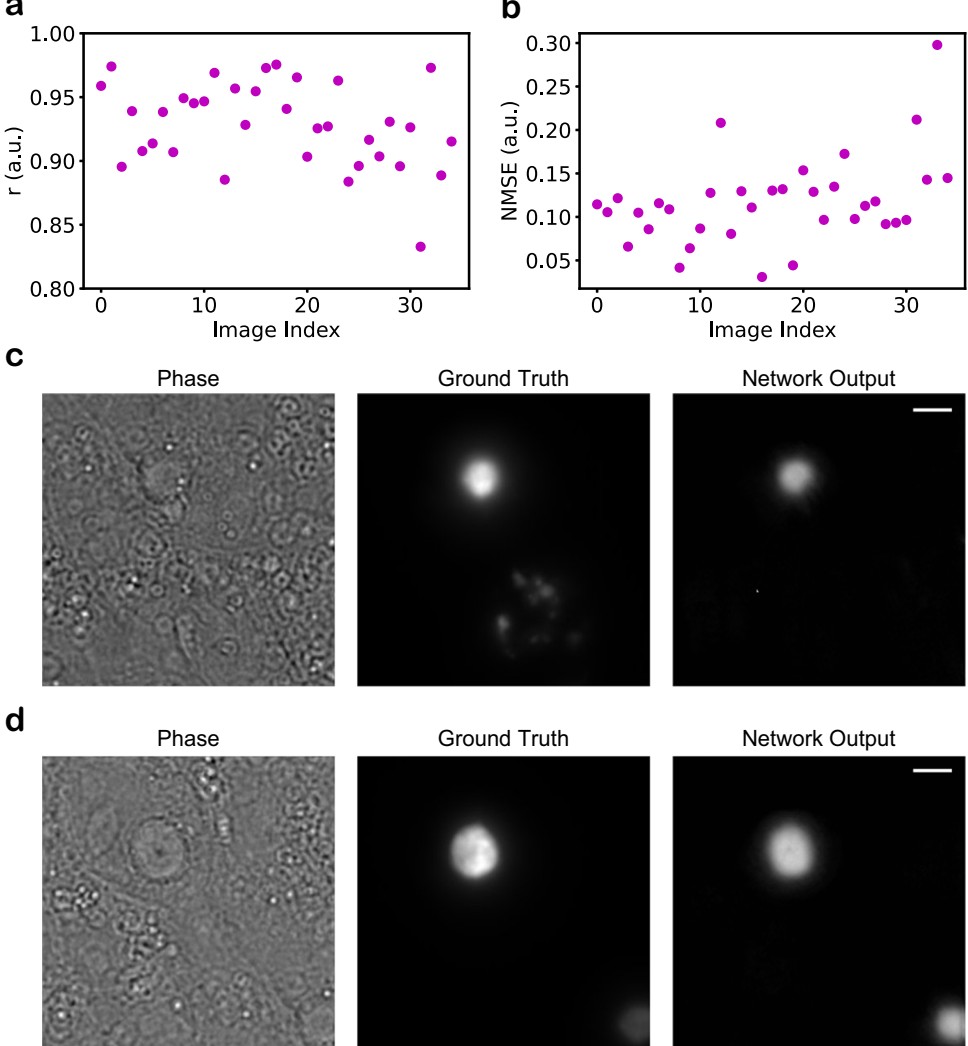

**Fig. 3 | Generalizability of LINA to a different cell line. a** Pearson correlation coefficient (*r*), computed only on the regions where there are aggregates. The metric is computed for the eight-plane-QPI pixel-regression model. **b** Normalized mean squared error, computed only on the regions where there are aggregates. The metric is computed for the eight-plane-QPI pixel-regression model. Both metrics show that LINA is able to accurately recognize aggregates expressed in a different cell line (HeLa). **c, d** Example images visualizing the model's performance on aggregates expressed in HeLa cells. **c** The example with the best *r*-value is shown. **d** The example with the lowest r value is shown. Scale bars: 5 μm.

with one, two, and four planes, respectively. Supplementary Fig. 7a quantifies and compares the performance of the eight-plane model to these three new models. We found that all three new configurations perform very well. We expected to possibly need transfer learning using a small set of new data to be able to identify aggregates correctly at much lower SNRs or for different cell lines, however, we found LINA to be highly robust to these changes, promising high generalizability to different kinds of imaging conditions. It is important to note that our QPI phase-retrieval algorithm yields better results with increasing numbers of planes. Therefore, it is possible that the model could perform even better if our training set had even more planes than eight. For our experiments, we settled on the standard eight planes while acquiring our dataset as a good trade-off between image quality and speed. This is because the eight planes are acquired simulta-neously using an image-splitting prism, whereas more planes would require a z-stack. Here, we tested the different numbers of planes after all eight were processed together into QPI images. Since the output image quality of our QPI method is dependent on the number of planes of the input[44], it could possibly be interesting in the future to also study the effect of having the QPI images themselves being generated by fewer planes, as well as the effect of varying the interplanar distance.

Our method to produce quantitative phase images[44] is widely adoptable; as previously mentioned, the phase information can be extracted using an algorithm from a z-stack of images. However, the simplest and most basic label-free technique is brightfield microscopy. How well would LINA perform on the simplest and most widely avail-able label-free technique? To answer this question, we trained pixel-regression models using only brightfield inputs, first with eight planes and then with a single plane (Supplementary Fig. 7b). We quantitatively compared both these models' performance with the model trained on eight-plane QPI images, and we found that both brightfield models perform very well on the test set. The eight-plane brightfield model yielded very similar results to the QPI model, with a slightly larger variance and the drawback of not having the possibility to extract the dry mass from the images.

To confirm LINA's generalizability and ease of use, we tested it on a completely different imaging setup with a different light dose, non-Koehler illumination, and a lower numerical aperture (NA) microscope objective (Supplementary Fig. 7c). We were able to identify aggregates from images acquired using this different setup and the 1-plane brightfield LINA model, without the need to rely on transfer learning. Next, we tested LINA on a commercial, confocal setup (Leica TCS SP8).

We used a similar magnification (×63 versus ×60) and NA (1.4 versus 1.2) objective, but the illumination, setup type (confocal versus wide-field), and detectors were completely different. Here, we acquired 8-plane stacks and used our QPI algorithm[44] to produce 8-plane QPI stacks. We tested LINA's performance using these QPI inputs (Supplementary Fig. 8a), as well as using the simplest possible inputs, 1-plane brightfield (Supplementary Fig. 8b), and both types of inputs yielded highly accurate network predictions. Based on these results, we expect LINA to work well for new users acquiring images at different conditions, perhaps needing some transfer learning, or as shown here, likely not even requiring it.

We further tested LINA on the same commercial setup, with a lower-magnification (×20 versus ×60) and NA (0.85 versus 1.2) objective (Supplementary Fig. 9). A lower-magnification objective means the FOV will be larger and, therefore, would enable users of our method to have a higher imaging throughput of cells containing aggregates. Since the network expects 352 pixels × 352 pixels inputs, we first cropped regions of that size around the aggregate images and used those as inputs to the network. We also directly tested the model on the 1024 pixels × 1024 pixels images. In both cases, the network performs very well and accurately identifies the aggregates. Here, again, we expected to possibly need transfer learning so that the network learns to expect smaller features (due to the lower magnification), yet the model works very well directly, illustrating the high generalizability and robustness of our method. These results using QPI inputs generated from images acquired on the commercial setup also demonstrate the ease of use of our QPI algorithm on a setup that is different from our own. For quantitative analysis of these images, further verification using technical calibration samples should first be done to ensure the accuracy of the phase retrieval, based on the optical properties of the setup that is used.

## Dataset size requirements

What is the minimal amount of data needed to successfully train a LINA model? To answer this question, we first computationally estimated this by taking increasingly sized samples from our complete dataset, which consists of over 1000 eight-plane phase and fluorescence image stacks. We calculated the mean aggregate area and circularity for each sample size, repeated the sampling 100,000 times, and took the average. We plotted the discrepancy from the ground truth standard deviation to indicate the sample size at which we obtain a similar distribution as that of the ground truth (Supplementary Fig. 11); here, we consider the ground truth standard deviation to be that of the entire dataset, since it leads to well-performing models and should be large enough to capture the diversity of the different kinds of aggregates. Supplementary Fig. 10 shows the morphological distribution of the entire dataset. The simulation showed a saturation at around 30–50% of the size of the complete dataset, indicating that there could be diminishing returns past this point.

We then aimed to answer this question experimentally by attempting to train models with varying amounts of data (10%, 20%, 30%, and 50% of the entire dataset). It is possible to train LINA models even with only 10% of our dataset (Supplementary Fig. 12). However, performance does degrade slightly for the models trained with lower amounts of data, except for the model trained with 50% of the data, which yields very similar results to our standard model trained with 90% of the data (the remaining 10% is used as the test set), in agreement with our simulation results. While it is possible to train a LINA model with just 10% of the data, we visually observed this model to not do as well as the 90% model on some test set images. It is crucial that the training and validation datasets capture the diversity of the subject of the data to ensure high generalizability and accuracy of the trained model, and, therefore, more data will very often lead to better results. However, care should be taken to only provide high-quality, useful data to the CNN, as low-quality data can actually confuse the network and lead to hallucinations[46]. The quality of the dataset is just as important as its quantity, making data curation a crucial step in successfully training a deep learning model.

## Automatic scanning and image acquisition

We enhanced our image acquisition throughput by implementing an automatic image-scanning method, called *xy*-scan mode, to scan an arbitrary number of fields of view (FOV) across the sample (Fig. 4a). The *xy*-scan mode permits us to set the step size and the number of steps in the *x* and *y* directions. The illumination source remains stationary while a piezo nano-positioning stage moves under the control of the software. This allows us to acquire images of thousands of FOVs, for cells expressing different Httex1 constructs, which our LINA models can then process to determine the presence or absence of aggregates (Fig. 4b).

Using the automatic xy-scan mode, we efficiently collected images of cells overexpressing mutant Httex1-39Q, Httex1-72Q, Httex1-ΔNt17-72Q, and Httex1-72Q-GFP, then processed them to determine which FOVs contain aggregates. To minimize the occurrence of false positives, we combined the predictions from multiple models. While this approach may result in a higher chance of false negatives, our priority is to ensure that the analysis results consider only true aggregates. The output of our model, which indicates the label-free images that contain aggregates, can then be used to extract their dry masses.

## Dry mass quantification and comparison between aggregates of different Httex1 constructs

The dry mass of a biological specimen can be extracted from a QPI image through a proportional relation (see the "Methods" section)[38]. For this relation, we used a constant value of 0.19 for the refractive increment α, as recommended in the literature, due to α having a highly narrow distribution[38,47]. The quality of the QPI images, e.g., in terms of resolution or SNR, determines the accuracy and precision of the dry mass measurement[38]. After the QPI image is translated into a dry mass density map, the dry mass of an aggregate can be measured by integrating over its area in the image (more details in Supplementary Fig. 13). To compare the dry mass of the different Httex1 constructs, we used the model outputs from the automatic xy-scan images to curate images with aggregates, producing segmentation masks that were then multiplied by the corresponding dry mass images. This allows for efficient and accurate measurement of the dry mass of the different kinds of aggregates (Fig. 4c).

Our findings indicate that there is no statistically significant difference between the mean dry masses of mutant Httex1-72Q and Httex1-ΔNt17-72Q. Both of these constructs have been shown to form inclusions with the same core-and-shell ultrastructure[14]. Thus, they likely have a similar mechanism of aggregation, which explains their similar dry masses. We observed that fewer aggregates were detected for Httex1-39Q (8% of automatically-scanned FOVs) compared to the other two constructs, which had similar rates (~12% of automatically-scanned FOVs). This is in agreement with the literature, as increasing polyQ repeat length is known to accelerate aggregation and toxicity[48–50]. However, interestingly, we found that Httex1-39Q, which lacks the core-and-shell ultrastructural arrangement[14], produced aggregates with a larger average area ($16.9 \pm 7.6 \, \mu m^2$), compared to Httex1-72Q ($11.5 \pm 4.7 \, \mu m^2$) and Httex1-ΔNt17-72Q ($12.7 \pm 4.9 \, \mu m^2$), and consequently a larger average dry mass. The difference in ultrastructure, and therefore the aggregation mechanism, leads to fewer aggregates being produced overall, yet at the same time, a fraction of the aggregates that are produced grow to larger final sizes.

We also compared the mean dry masses of the three unlabeled constructs with that of the labeled one (Httex1-72Q-GFP) and found around a two-fold increase in the mean dry mass of the labeled construct compared to the two constructs with the same polyQ repeat length, as well as a 1.5-fold increase compared to Httex1-39Q. Similar to Httex-39Q, Httex1-72Q-GFP aggregates have been shown to lack the

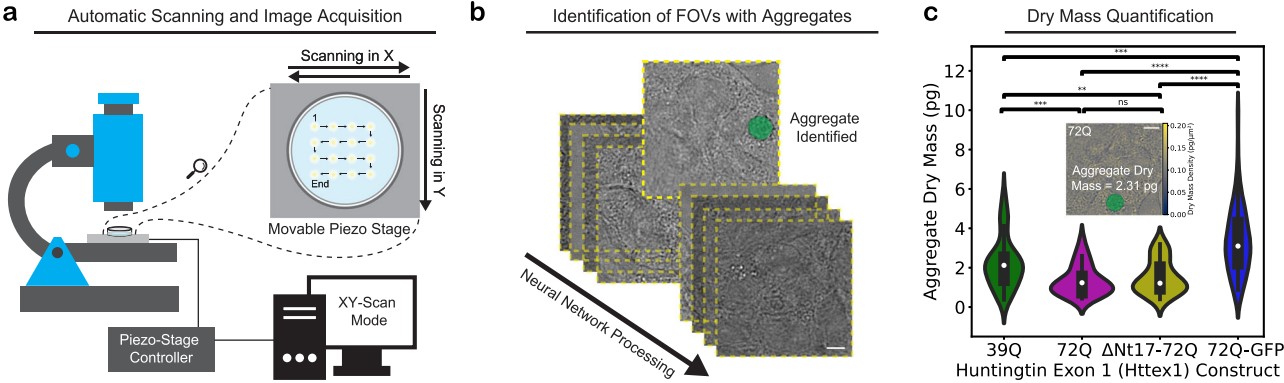

**Fig. 4 | Automatic image acquisition, identification, and dry mass quantification of different kinds of label-free Httex1 aggregates. a** A motorized piezo nano-positioning stage is controlled using software to scan the sample in the *x* and *y* directions and collect images at various fields of view (FOVs) for different label-free constructs of Httex1 (39Q, 72Q, ΔNt17-72Q) and a GFP-labeled construct (72Q-GFP). **b** The images are post-processed and our trained model is used to identify FOVs that contain aggregates. **c** The dry mass of aggregates produced by different constructs of Httex1 is extracted from the quantitative phase images, as shown in the inset image (*n* = 39 for 39Q, *n* = 29 for 72Q, *n* = 30 for ΔNt17-72Q, and *n* = 64 for 72Q-GFP). ΔNt17-72Q and 72Q had similar dry mass distributions (unpaired, two-sided *t*-test resulted in a *p*-value of 0.40), however, both label-free constructs had smaller dry masses on average than 39Q (unpaired, two-sided *t*-test *p*-values of 0.0008 and 0.0085 for 72Q and ΔNt17-72Q, respectively), and all three had smaller dry masses than the labeled 72Q-GFP (unpaired, two-sided *t*-test *p*-value = 0.00012 for 39Q, 2.8e−12 for 72Q, and 2.1e−10 for ΔNt17-72Q). ns: *p* > 0.05, **$p \leq 0.01$, ***$p \leq 0.001$, ****$p \leq 0.0001$. White dots represent the medians, thick bars represent the interquartile ranges, and thin lines represent 1.5× the interquartile ranges. Scale bars: 5 μm.

core-and-shell ultrastructural arrangement[14], which for both constructs leads to increasing the size of the formed aggregates. The considerable increase in the size of Httex1-72Q-GFP aggregates is likely due to a mixture of the large size of the GFP tag and its effect on the size of the fibrils, the difference in ultrastructure, and the difference in properties (e.g., proteome composition) that it is known to cause, supporting previous studies' results and demonstrating why such label-free methods are needed.

### Live, label-free identification and analysis of an aggregate as it forms

Capturing the growth dynamics of NDD-associated protein aggregation is crucial to deciphering the role of different stages and intermediates on the pathology formation pathways in the pathogenesis of NDDs and potentially identifying new targets for therapeutic intervention. For example, if the aggregate growth rate is controlled by a particular cellular mechanism, drugs that target this mechanism may slow or stop the disease development and progression. Furthermore, a better understanding of the characteristics of different aggregates on the pathway to pathological inclusions should facilitate the development of new molecular probes and diagnostic tools for monitoring early oligomerization events that are thought to contribute to and correlate with early stages of disease development. LINA enables the live identification, imaging, and analysis of native aggregates as an important step toward reaching these goals.

Figure 5a shows our LINA pixel-regression model's performance on dynamic, live-cell, time-lapse images, taken every 2 min, of Httex1-72Q-GFP in a HEK cell, starting from a diffuse protein state and aggregating over time. LINA can distinguish between the diffuse protein and the aggregated state, as it only starts to predict an aggregate at the time-point where the aggregate begins to form, as verified visually from the ground-truth fluorescence images. It is able to correctly detect the aggregate as it grows in size from then on, and as it moves along different subcellular localizations. Figure 5b illustrates the normalized mean intensity of the output images over time. As the aggregate grows and more pixels are predicted as parts of an aggregate, the mean intensity of the output image increases. The mean intensity follows the expected sigmoidal behavior and can be divided into three regimes. First, for diffuse protein, the prediction intensities are very low, which shows successful predictions that there are no aggregates. Then, for aggregate formation, the intensity starts

increasing gradually while the aggregate grows, until the third regime is reached, where the intensity stabilizes into a steady state and the aggregate stops growing.

The dry mass of aggregates can be extracted from the phase images and the segmentation masks of the network output images in the same fashion as described previously for the fixed cells. Furthermore, the area can also be extracted from the mask of the identified aggregate. Both the dry mass and area are plotted in Fig. 5c, and they are highly correlated to each other and to the normalized mean intensity. Once again, they can be split into three regimes: diffuse protein, aggregate formation, and stabilizing to a steady state. The ability to measure the dry mass and area of living aggregates as they grow opens up many possibilities for dynamics studies.

### Discussion

LINA is a label-free method for virtually labeling and identifying protein aggregates using a trained deep-learning CNN, enabling more accurate analysis of the native aggregates, and avoiding label-induced alterations. We have validated LINA quantitatively, showed its applicability on different constructs of Httex1, and verified the models' consistently-accurate performance over various imaging conditions, including varying SNR, cell line, and input type, illustrating the universality of our method. Furthermore, LINA is fully-compatible with long-term, live-cell imaging, circumventing the issues of photobleaching and phototoxicity, and can be used to identify aggregates and analyze their quantitative parameters (e.g., area, dry mass, morphology, and interaction dynamics). In support of prior studies[14,16,17,51], our characterization and comparison of the dry masses of unlabeled and labeled aggregates further demonstrate the need for using label-free methods to study protein aggregation and inclusion formation in NDDs. We found that mutant Httex1 with a truncated Nt17 domain and a polyQ repeat length of 72Q forms aggregates of the same dry mass on average as the construct with the same polyQ repeat length but maintaining the Nt17 domain. We found that the aggregates formed by Httex1 with a shorter polyQ repeat length of 39Q, which lack the core-and-shell ultrastructural arrangement of the two other constructs, are larger and heavier on average, despite a reduction in the number of formed aggregates. This is likely due to the different ultrastructure and aggregation mechanisms of Httex1-39Q. Further analysis and comparisons of the characteristics of different label-free Httex1 constructs will be possible using LINA, which could lead to new discoveries into

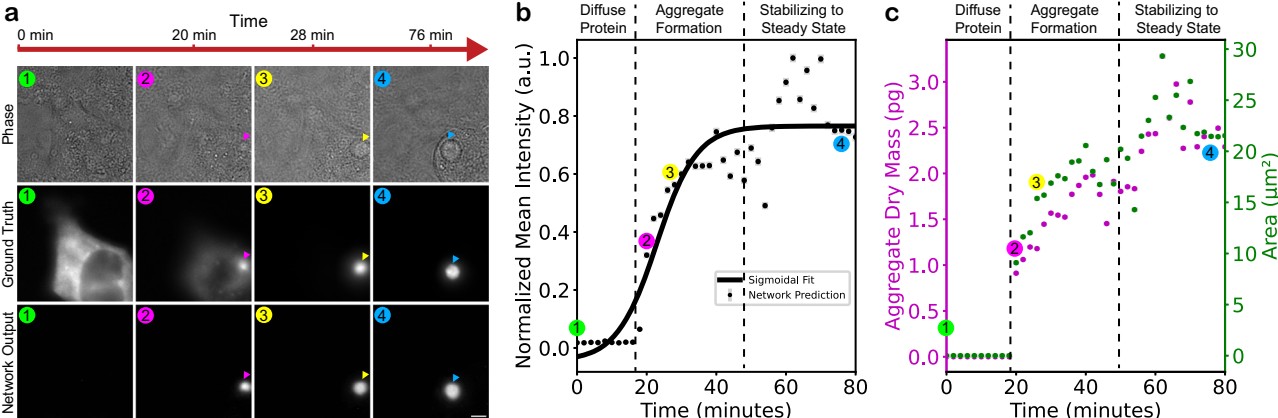

**Fig. 5 | Live, label-free identification and analysis of Httex1 protein aggregation. a** Time-lapse images, acquired every 2 min, of Httex1-72Q-GFP in a HEK cell, starting from a diffuse protein state and aggregating over time. LINA is able to distinguish between the diffuse protein and the aggregated state, correctly predicting the aggregate as it grows in size and moves along different subcellular localizations. Scale bar: 5 μm. **b** Normalized mean intensity of the network output images as the aggregate grows, following a three-regime sigmoidal behavior. **c** Dry mass and area changes extracted from the network output images, split into the same three regimes. **b**, **c** Error bars represent the standard deviation in each image. The data shown is for one field of view as the aggregate forms.

the pathological mechanisms of Huntington's disease and uncover possible therapeutic targets.

Nowadays, artificial intelligence (AI), particularly machine learning, is extensively being used in microscopy for various purposes, including denoising, image-to-image translation, segmentation, classification, and even medical diagnosis[39–42,52–57]. There have also been developments aiming to ease the adoption of these models and democratize their use[58–60]. As with any new technology, there can be mistrust and a reluctance to rely on it, especially when it is a "black box" that is difficult to interpret—fortunately, there has been considerable progress in deciphering how deep learning models make their predictions[46]. In the case of AI-enhanced microscopy, concern can be warranted because of the field's fundamental nature and its being a basis for biomedical discoveries, which makes it truly important for any results to be unequivocally true, as this affects future studies and research directions which, depending on their success, can either facilitate or delay the treatment of devastating illnesses. This is especially the case when it comes to unsupervised learning using generative models, such as those using a generative adversarial network (GAN). GANs create highly plausible outputs that can be hard to distinguish from real data, and it can be difficult to determine if biological phenomena identified by such a model are true or a hallucination. To mitigate such concerns, it is important to validate the AI model as much as possible and to discuss the kind of problem that it excels at solving, as well as the model's limitations. That is why we validated our models as much as possible, quantitatively, on different protein constructs and at different conditions.

The detection limit of LINA depends on several factors, notably the size distribution of the training dataset and the limitations in the images resulting from the microscope (contrast, SNR, resolution) and the sample (transparency, refractive index heterogeneity). In our training dataset, we focused on late-stage aggregates, making sure to include images of aggregates of varying morphologies and sizes, such that the network can be as general as possible. We show that LINA works well for varying image SNRs, aggregate sizes and shapes, protein constructs, cell lines, and in live-cell imaging conditions. The current version of LINA can detect aggregates with areas as small as 3 μm². To further enhance LINA's capabilities for kinetics studies, we aim in future work to expand the training dataset to include early and intermediate-stage aggregates, which would enable the detection of even smaller aggregates than what is currently possible. This can be accomplished by re-training a new model on the expanded dataset, or through transfer learning on a dataset consisting of the earlier-stage

aggregates. Low-affinity binders[61] that could better capture the diversity of misfolded protein aggregates could have great potential to extend our method to enable the identification and label-free imaging of different aggregation states on the pathway to inclusion formation.

We believe that our method has significant potential for characterizing protein aggregates and inclusions associated with other NDDs. This could be achieved by following a similar approach of training a CNN from scratch on a new dataset, or by simply performing transfer learning on our trained model with the new data. Methods to increase the throughput of the data acquisition, such as the automatic XY-scan mode we use here, are useful for collecting new data for transfer learning or re-training. Another possibility would be to image at a lower magnification, increasing the yield from one image, which we showed is a possibility using our method (Supplementary Fig. 9). However, this comes at a cost of resolution and information, which will likely complicate the identification of small aggregates from transmitted-light images, which could be particularly problematic for live-cell imaging of aggregates as they form. Our method has the advantage of not requiring manual annotations, as the training labels are directly obtained from the fluorescence images. Therefore, researchers who are interested in extending our approach using new data only need access to a microscope capable of imaging in a label-free modality and any kind of fluorescence microscopy. Additionally, LINA could potentially be extended in the future to virtually label aggregates from electron microscopy images. LINA paves the way for studying the molecular interaction dynamics of protein aggregates and the evaluation of the cellular state in cells containing aggregates, without the need to rely on labels. It can be extended to virtually label other cellular components, such as organelles or cytoskeletal proteins, which are known to interact with NDD-associated protein inclusions, and several studies have already shown their capability to be virtually labeled[39,40,62]. Moreover, LINA can be further developed to classify the cellular state[39] of label-free cells containing aggregates to elucidate the mechanisms that disrupt cellular viability or introduce toxicity. We also envisage several possibilities for LINA to be correlated with other imaging techniques, opening up new avenues for correlative microscopy of neurodegenerative diseases.

## Methods
### Sample preparation
HEK 293 cells were cultured at 37 °C and 5% CO₂ using DMEM high glucose without phenol red (Gibco, Thermo Fisher Scientific), supplemented with 10% fetal bovine serum, 1% penicillin–streptomycin

and 4 mM L-glutamine (all three from Gibco, Thermo Fisher Scientific). Cells were plated at a density of 120,000 per dish either on high precision #1.5 25 mm coverslips (Marienfeld) or on FluoroDish Sterile Culture Dishes 35, 23 mm well (World Precision Instruments), coated with fibronectin. Cells were transfected one day after plating using polyethylenimine (PEI) transfection. 2 µg of DNA were mixed in 100 µl of OptiMEM Reduced-Serum Medium (Life Technologies), 6 µl of PEI were mixed in 100 µl of OptiMEM, and then both mixtures are mixed and incubated for 5 min at room temperature (RT), then added dropwise and carefully distributed over the cells. Cells were then returned to the incubator and left either for 48 h before fixation or for shorter durations before live-cell imaging followed by fixation at 48 h post-transfection. For fixation, cells were washed twice with PBS pH 7.4 (1X) (Life Technologies, Switzerland) and fixed in 3.7% formaldehyde (Sigma-Aldrich, Switzerland) in PBS (PFA) for 15 min at room temperature (RT). Cells were then washed in PBS and then mounted in PBS.

For immunocytochemistry, after a blocking step with 3% BSA (Sigma-Aldrich, Switzerland) diluted in 0.1% Triton X-100 (Applichem, Germany) in PBS (PBST) for 30 min at RT, cells were incubated with the primary antibody (anti-Htt raised against the Proline-rich domain (PRD) (MAB5492, Millipore) at a dilution of 1/500 in PBST for 2 h at RT. Cells were then rinsed five times in PBST and incubated for 1 h at RT with the secondary donkey anti-mouse Alexa488 (Life Technologies, Switzerland) used at a dilution of 1/800 in PBST. Cells were then washed five times in PBST, and finally washed once in double-distilled H$_2$O, before being mounted in PBS.

### Data acquisition and processing
Most imaging was performed with a custom-built microscope (Supplementary Fig. 14) equipped with a temperature and CO$_2$-controlled incubator for live cell imaging, as described in previous work from our group[44]. Live-cell imaging was performed in DMEM without phenol red at 37 °C and 5% CO$_2$. The microscope is controlled using custom LabVIEW software. For fluorescence imaging, a 120 mW, 488 nm laser (iBeam smart, Toptica), is focused into the back focal plane of an Olympus UPLSAPO 60XW 1.2 NA objective for wide-field epi-fluorescence illumination. The fluorescence light was filtered using a combination of a dichroic mirror (zt405/488/532/640/730rpc, Chroma) and an emission filter. For phase imaging, we used the white-light Koehler illumination module of a Zeiss Axiovert 100 M microscope equipped with a halogen lamp to collect brightfield images which are later processed into QPI images. The detection path is arranged as a sequence of four 2-f configurations to provide image–object space telecentricity. The image splitter placed behind the last lens directs the light into eight images, which are registered by two synchronized sCMOS cameras (ORCA Flash 4.0, Hamamatsu; back-projected pixel size of 111 nm; interplanar distance of 350 nm). For translating the sample, the microscope is equipped with piezoLEGS stage (3-PT-60-F2,5/5) and Motion-Commander-Piezo controller (Nanos Instruments GmbH). While collecting our dataset, we acquired any FOV that had an aggregate, not filtering for particular sizes or shapes, to have a dataset that is as general as possible. For all imaging experiments and data collection, multiple samples from at least three independent experiments were used for data acquisition. For confocal imaging, we used a Leica TCS SP8 with two objectives (×20 magnification, 0.85 NA and ×63 magnification, 1.4 NA). We used the same interplanar distance (350 nm) as for our commercial setup, when collecting z-stacks to be used as inputs to our eight-plane models.

We used custom MATLAB (R2021a) (Mathworks) scripts (available here) to retrieve the phase information from the brightfield images and produce quantitative phase images. These scripts are also used for pixel registration in the eight z-planes for both phase and fluorescence images. The images are cropped to a size of 352 pixels × 352 pixels.

We used Fiji[63] (v2.9.0) scripts to produce maximum z-projection fluorescence images, to segment these images using Otsu thresholding and produce the labels for pixel classification, to prepare the color-coded maximum z-projection phase image shown in Fig. 2a, and for the image processing in Fig. 2b. Fiji was also used to segment network predictions and produce masks which are used to measure the area, circularity or dry mass of aggregates.

Python (v3.7) was used for data analysis and plotting. This includes the quantitative metric calculations and cropping the regions of interest (ROIs), i.e. the aggregate regions within the images. This was done using Otsu thresholding to produce a mask from the ground truth image, which is then used to crop the ROIs. The Pearson correlation coefficient was calculated using the function 'Pearsonr' in the Scipy library (v1.7.3), and the MSE was calculated using the mean_squared_error function in the Scikit-learn library (v1.0.2), then normalized by the square of the sum of the label images, as defined in the literature[64].

### Neural network training
Our models are trained on a deep CNN with a U-Net[45] architecture (Supplementary Fig. 15). Compared to the original architecture, we reduced the number of feature maps by a factor of 4 which led to a reduction in the trainable parameters by a factor of 15. This notably reduces GPU memory usage and training time, while still enabling excellent performance. We used TensorFlow (v2.8.0) and Keras (v2.8.0) to build our network, and training was done on a workstation equipped with an NVIDIA GeForce RTX 3090 GPU. We used the adaptive moment estimation (Adam) optimizer with a learning rate of 1e−4 and a mean squared error loss function. Before training our models, we normalize both the phase and fluorescence images by rescaling each image to be between 0 and 1. We used 10% of our dataset as the test set and split the rest of the dataset into training and validation sets with 20% being used for validation. We used the 'EarlyStopping' (on the validation loss) and 'ModelCheckpoint' callbacks to avoid overfitting and to save the best, most general models.

### Dry mass quantification
Supplementary Fig. 13 summarizes the dry mass extraction process. To map between phase information and dry mass, we use the following equation[38]:

$$\sigma(x,y) = \frac{\lambda}{2\pi\alpha}\phi(x,y) \tag{1}$$

where $\sigma$ and $\phi$ are the dry mass and the phase, respectively, at the location defined by the $x$ and $y$ coordinates, $\lambda$ is the wavelength of the illumination (here chosen as the average wavelength of our white-light source), and $\alpha$ is the refractive increment, treated as a constant value of 0.19 µm$^3$ pg$^{-1}$, as recommended in the literature, due to α having a highly narrow distribution[38,47].

### Reporting summary
Further information on research design is available in the Nature Portfolio Reporting Summary linked to this article.

## Data availability
The training and test datasets are available on Zenodo[65]. Source data are provided with this paper.

## Code availability
The Python codes for training a model, using a pre-trained model, and for transfer learning on a pre-trained model are available on GitHub[66] at: https://github.com/kibb/LINA/.

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

## Acknowledgements

We thank Samuel M. Leitao for testing the method on a different imaging setup and Anne-Laure Mahul-Mellier for the helpful discussions. This work was financially supported by the SV-iPhD program at EPFL. A.R. and L.F. were supported by funding through the Swiss National Science Foundation (SNSF) through the National Centre of Competence in Research Bio-Inspired Materials and SNF grant CRSII5_193740. K.S.G. acknowledges the TU Delft AI Labs program for support.

## Author contributions

K.S.G. conceived the idea and initiated the project, which was driven by the observations of N.R.; K.A.I., H.A.L., and A.R. further developed the idea and designed the experiments; K.A.I., L.F., and N.R. prepared the samples; K.A.I. performed the experiments, acquired, analyzed, and visualized the data, developed and optimized the neural network training pipeline, validated the trained models, and developed the pipeline to extract the dry mass and area measurements; K.A.I. and H.A.L. wrote the paper, with input from all authors. H.A.L. and A.R. supervised the project; All authors discussed the results and commented on the manuscript.

## Competing interests

H.A.L. has received funding from the industry to support research on neurodegenerative diseases, including from Merck Serono, UCB, and Abbvie. These companies had no specific role in the conceptualization, preparation, and decision to publish this work. H.A.L. is also the co-founder and Chief Scientific Officer of ND BioSciences SA, a company that develops diagnostics and treatments for neurodegenerative diseases based on platforms that reproduce the complexity and diversity of proteins implicated in neurodegenerative diseases and their pathologies. All remaining authors declare no competing interests.
