## [Peer review file · Nature Communications]

REVIEWER COMMENTS

Reviewer #1 (Remarks to the Author):

In this manuscript, Khalid A. Ibrahim, Kristin S. Größmayer, Nathan Riguet, Lely Feletti, Hilal A. Lashuel and Aleksandra Radenovic present their work on using a U-Net-like deep neural network for identifying protein aggregates in phase contrast images. While there is no novelty or originality in the architecture and training of the neural network model, the authors have demonstrated its generalizability and robustness in label-free identification of Httex1 aggregates through a series of follow-up experiments.

I believe the value of this work lies in its high robustness and generalizability. In this context, the lack of novelty in the machine learning aspect is more of a virtue than a drawback. Using an easily applicable network and training strategy can ensure reproducibility by other researchers. Moreover, the level of generalizability and robustness implies that there is limited room for improvement in this type of classification and regression tasks. Considering the rapid changes in the neural network field, the demonstrated robustness and generalizability may be more valuable than a novel breakthrough in neural network model design with marginally improved performance.

My comments and suggestions primarily focus on the broad generalizability and robustness of the work, with the aim of benefiting researchers who require quantification of protein aggregation using widefield images of model cell systems. The authors need not address all of my comments literally; if the study is actually aimed at more specific protein aggregations and readers, it would be sufficient to provide a detailed description of the network's applicability and limitations for that purpose.

Comments

1. Regarding model validation (Fig. 2c): The pixel regression network employed SSIM and Pearson correlation, but these two metrics may not be suitable for validation in this study. Pearson correlation generally underperforms in image comparison tasks, while SSIM aims to mimic human perception in image comparison. Consequently, when most of the output images consist of noise or background, and only a small ROI contains the actual signal (as is the case in this study), these metrics may produce large discrepancies in the results. Therefore, I suggest the following:
 - a. Present normalized MSE: Since MSE was utilized as the training loss, it would be beneficial to display the results in a human-comprehensible format for validation purposes.
 - b. Employ metrics with better pixel-wise accuracy representation: Generally, Dice loss and its variants are more suitable for this purpose. However, the authors can explore alternative metrics with appropriate justifications. If the authors prefer to maintain SSIM and Pearson correlation, it is advisable to crop images around the ROI or apply a certain threshold. The specific approach for generating these metrics should be described in the methodology. The same cropping or thresholding technique can also be applied when representing the MSE numbers (Comment 1-a) since the background MSE might be somewhat significant due to the large area it occupies.
 - c. Complement metric representation with intensity comparisons: In addition to comparing the pixel-wise similarity of ground-truth and network-output images using metrics, it is essential to compare the total intensity of ground-truth fluorescence images and network-output images for aggregations. Without this validation, the network cannot be effectively used for quantification purposes in kinetics studies, as its dry mass cannot be quantified without supplementary dry mass information (e.g., QPI-based dry mass images in this study).
2. Regarding Generalizability: Although the training and validation of the network are based on Httex1 aggregates and their variants, the network's robustness and consistent performance across different Httex1 variants suggest that it may be applicable to various other protein aggregates in general. This implies that the network might have learned features unique to aggregates, distinguishing them from cellular component features. It appears that the authors intended to emphasize this aspect, as evidenced by the manuscript's title. However, there are a few concerns to address:

a. **Circularity:** The images presented in the main figures and the distribution of morphology (Supplementary Fig. 6) show that aggregates possess highly circular morphologies. However, fibrillar morphology is frequently observed in NDD studies [1]. Moreover, more diverse morphologies can arise when considering that aggregates can form alongside interacting cellular organelles [2]. It would be beneficial for the authors to address this issue. One possible approach is simulating non-circular morphologies in their test/validation images. Given that the authors already have a well-performing classification network, they could use it to determine the aggregate's boundary and introduce distortions in the subregions to generate non-circular aggregation images. Alternatively, the authors could add a note explaining why circularity does not impact generalizability.

[1] Boatz, J. C., Piretra, T., Lasorsa, A., Matlahov, I., Conway, J. F., van der Wel, P. C. A. (2020). Protofilament Structure and Supramolecular Polymorphism of Aggregated Mutant Huntingtin Exon 1. *Journal of Molecular Biology*

[2] Zhou, Y., Peskett, T. R., Landles, C., Warner, J. B., Sathasivam, K., Smith, E. J., Chen, S., Wetzel, R., Lashuel, H. A., Bates, G. P., Saibil, H. R. (2021). Correlative light and electron microscopy suggests that mutant huntingtin dysregulates the endolysosomal pathway in presymptomatic Huntington's disease. *Acta Neuropathologica Communications*

b. **Optical properties of aggregation:** The authors have demonstrated that the trained network is robust across different variants of Httex1 aggregations, and even outputs from brightfield images are quite reliable. When considering protein aggregates from other proteins, differences will be manifested in their refractive indices, leading to phase differences in QPI or intensity variations in brightfield images. Consequently, the contrast between protein aggregation and cellular background images becomes critical. If the authors can present a working range of SNR concerning protein aggregation (signal) to cellular background (noise) for its robustness, it could justify the network's general applicability without conducting additional experiments on diverse protein aggregations. One approach could be to add noise specific to the protein aggregation region or blur the protein aggregation region with cellular images.

c. **Sharing the training weights (i.e., the pre-trained model)** is highly recommended. The generalizability claims are based on the trained network, rather than the network architecture or training strategy. In my opinion, if the trained model is not publicly available, the generalizability loses much of its value. However, I could not find any mention of the availability of the pre-trained model in the current version of the manuscript.

d. **Specification of the detection limit:** Supplementary Fig. 6 reveals that the training data do not include small-sized aggregates. It would be beneficial to understand the extent to which the network can identify and quantify protein aggregations for use in kinetic studies. Additionally, it would be helpful to provide information on false positives, such as presenting representative outputs for cell images without aggregation as examples.

The following suggestions may not be critical for this study and could be beyond its scope. However, in my humble opinion, these recommendations could enhance the study's value for the scientific community and related research fields.

Suggestions

1. Since QPI might not be readily available in many biology labs, it could be interesting to examine the network's performance on phase-contrast and DIC images, which are more commonly accessible. Furthermore, if the network has indeed learned to distinguish protein aggregation features from cellular features, it is possible that it may also work on EM images without additional training.

2. **An online server:** Although machine learning and neural networks are becoming increasingly popular, many researchers still lack the resources and expertise to utilize published models effectively. To maximize the benefits for both authors and the scientific community, it would be advantageous to create an online server where users can simply submit an image and receive output from the pre-trained model. Given the generalizability and robustness of this network, it has the potential to become a standard label-free quantification method. By including a note encouraging users to cite the relevant paper when using this resource, the authors could generate significant impact and recognition for their work. Imagine the number of citations that could be received if every time someone used

SDS-PAGE products or employed the theory of relativity in their research, they were asked to cite the original electrophoresis or the relativity theory paper. Unfortunately, Einstein and the developers of electrophoresis forgot that notes, even though they received Nobel prizes. But you can put that note! Though a Nobel prize is not guaranteed.

3. Heterogeneity. This point is an extension of Comment 2-b, but addressing heterogeneity could be a significant undertaking, so it is presented as a separate suggestion. Structural variance, particularly when it involves structural heterogeneity, can result in inhomogeneous optical properties [3]. Therefore, it would be worthwhile to explore whether the network performs well with these heterogeneous aggregations. Once again, this could potentially be achieved by creating chimeric aggregation images from the training images, allowing for a more comprehensive assessment of the network's performance.

[3] Meng, F., Yoo, J., Sung, H. (2022). Single-molecule fluorescence imaging and deep learning reveal highly heterogeneous aggregation of amyloid- β 42. Proceedings of the National Academy of Sciences

In conclusion, I believe the paper is nearly complete as it stands, particularly in terms of the experimental validation of the trained network. The authors have demonstrated careful and thorough addressing of every aspect to ensure the network's generalizability within the current scope of the study. However, I believe this work has the potential for even broader generalizability. If this appeals to the authors, I hope my other comments will be helpful in revising the manuscript accordingly or future follow-up studies. Please note that not all of my comments are of critical importance (except for 1-a and 1-b), and they should be used as guidance in line with the authors' revision direction. These points will be reiterated to the editor.

Reviewer #2 (Remarks to the Author):

Ibrahlim and coworkers develop a CNN based approach to identify and quantify aggregates in bright field images of Huntington's disease cellular models. The authors perform appropriate training and validation of LINA (the proposed tool) demonstrating that it performs well on unseen samples. The idea behind the approach is to enable fast, accurate, broadly-applicable ultra-structural characterisation of aggregates implicated in neurodegenerative diseases. To address this mission, I have a number of important concerns that need to be addressed before I can recommend this for publication:

1, in their literature review, the authors do not mention an important class of in situ fluorescent reporters that are minimally invasive, broadly applicable to small and large aggregates of different proteins from different samples, and bright enough to enable super resolution imaging. These reporters range from epitope and conformation specific antibodies / nanobodies to conformation specific, low Kd binders (such as ThX, YOYO, etc.). Why is a label free approach, that is less accurate in its quantification than a much more broadly applicable fluorescence based approach and as quick, be favoured?

2, aggregates, at least from other proteins such as tau, aSyn, and abeta, exhibit different conformations in different diseases, under different recombination conditions, and between different samples. Given that cellular models do not recapitulate human disease pathology, how well is the proposed approach suited to characterise aggregates from disease relevant samples (bio fluids, post mortem brains, and human-derived organoids) and different proteins and why has this not been shown.

3, the aim behind the proposed approach is to enable better understanding of the role of aggregation in disease progression, however, the authors didn't demonstrate any new functional insights from the application of LINA to their cellular models beyond measurement of dry mass over time.

4, the authors use custom built instrumentation and a LabVIEW software for data acquisition

automation. Can the workflow be demonstrated on a commercial microscope and an open source data acquisition tool to demonstrate general use. The same could be also though for the training model - which having looked at the having available source code - is not sufficiently annotated or packaged into a GUI-based software to be used by biologists who might be interested in applying the tool.

Reviewer #3 (Remarks to the Author):

In their paper "Label-free Identification of Protein Aggregates Using Deep Learning" by Ibrahim et al., the authors present an approach (LINA) to study protein aggregation without using fluorescence labeling other than during the training of a machine learning algorithm. The authors are using quantitative phase reconstruction from multiple plane brightfield imaging to obtain both virtual staining used for segmentation of aggregate and dry mass quantification of these lasts. The authors claim the generalizability of their machine learning approach over different variable such as the type of protein, cell type, live/fixed, type of imaging (BF of phase), microscope variation and noise. The overall paper is well written and organized with convincing results which are mainly impacting the biological community and rely on well-known experimental tools and algorithm strategies. I suggest some improvement on the paper to make it more attractive and improve the spread in a larger community.

- My major concern is regarding the ambiguity of fluorescence labeling. Indeed, as very clearly stated in the paper, the use of fluorescence labelling may perturbate aggregation. However, to train the algorithm, fluorescence labelling has to be performed. In the study case proposed by the authors, the cell line and labelling strategy are chosen not to perturbate the aggregation but how to generalize this to other protein aggregations since live imaging for dynamic aggregation/de-aggregation study is researched? Does immunohistochemistry perform as good as GFP based transfection for the training since it might be the general procedure? This is demonstrated with a training on transfected cells applied on histochemistry stained cells but does the opposite still work?

- My second concern is on the observation scale. Indeed, clusters are large ($>3\mu\text{m}$) and the numerical aperture and magnification used are important. Is it a requirement (for accurate phase reconstruction for ex.)? Indeed, the authors are using a automatized XY-scan to have large FoV but it would be more interesting to have the same study directly with low mag and NA objectives. I would recommend a test at low magnification since it could greatly enhance the spread of this technique with a huge enhancement in the acquisition speed.

- The use of phase rather than just intensity is giving a very little gain, this has to be stated in the abstract by removing the requirement of using phase information until the discussion on the dry mass. This will strengthen the generalizability of the LINA approach.

- Since the reconstruction is 3D for both phase and fluorescence, why no 3D study has been performed? This has to be done or at least discussed in term of memory size, speed, gain in segmentation...

- I would like some details on the live study. The model used is trained on fixed cells. But what was the moment of aggregation (final I assume). Will it help to have a more diverse dataset with all stages of aggregation?

- On the model transfer to other cell lines (ex. HeLa), I want to see images and not just SSIM curve for both the cells that are working well (high SSIM) and bad (lower SSIM). Does the virtual staining performance linked to a cell shape in particular? I also recommend to put it back in the main part of the paper the whole study since it is a major result for the algorithm generalizability.

- On the acquisition platform: I would like a SI image of the setup in order to have a complete overview of the method.

- On the data processing and AI: a more detailed section in the SI could greatly enhance the paper. With notably

- o a discussing on the presence or absence of overfitting and how it is checked.
- o a scheme of the U-net used.
- o a precision of the dimension of the input/output.
- o a global data processing pipe from the initial raw data down to the characterization of the clustering size and dry mass.
- I would prefer to see the discussion part on the minimal amount of data need to train LINA in the result part. It would fluidify the message.

Label-free Identification of Protein Aggregates Using Deep Learning

Khalid A. Ibrahim, Kristin S. Größmayer, Nathan Riguet, Lely Feletti, Hilal A. Lashuel, and Aleksandra Radenovic

Manuscript #: NCOMMS-23-17657

Response to Reviewers

We are grateful to all three reviewers for their careful reading of the manuscript and their constructive feedback and recommendations, all of which have contributed significantly to improving the quality and impact of the manuscript.

Both reviewers #1 and #3 spoke highly about the thoroughness, relevance, and impact of the paper, and recommended its publication. They clearly mentioned that the points and requests for clarifications they made are aimed at improving the generalizability of the paper and to make it more attractive for the larger community.

Reviewer #1: In conclusion, I believe the paper is nearly complete as it stands, particularly in terms of the experimental validation of the trained network. The authors have demonstrated careful and thorough addressing of every aspect to ensure the network's generalizability within the current scope of the study.

Reviewer #3: The overall paper is well written and organized with convincing results which are mainly impacting the biological community and rely on well-known experimental tools and algorithm strategies. I suggest some improvement on the paper to make it more attractive and improve the spread in a larger community.

Reviewer #2 comments relate mainly to three points; 1) highlighting the advantage of the label-free approach compared to the use of amyloid dyes or antibodies to image inclusion formation; 2) relevance of the models used in relation to the complexity and diversity of protein aggregates in the brain; 3) showcasing the generalizability of LINA on a commercial microscope and clarifying the advantages and potential of our technique.

We have made every effort to experimentally address their questions and provide thorough responses to their requests for clarifications and inquiries. Please find enclosed a point-by-point response to all their questions and inquiries.

Reviewer comments are reproduced in black, our answers are in blue, and changes to the manuscript are in green. *References to the manuscript text are in green italics.* Citations correspond to a reference list at the end of this document. Please note that all line and page numbers refer to the revised manuscript, unless indicated otherwise.

Reviewer #1:

In this manuscript, Khalid A. Ibrahim, Kristin S. Größmayer, Nathan Riguet, Lely Feletti, Hilal A. Lashuel and Aleksandra Radenovic present their work on using a U-Net-like deep neural network for identifying protein aggregates in phase contrast images. While there is no novelty or originality in the architecture and training of the neural network model, **the authors have demonstrated its generalizability and robustness in label-free identification of Httex1 aggregates through a series of follow-up experiments.**

I believe the value of this work lies in its high robustness and generalizability. In this context, the lack of novelty in the machine learning aspect is more of a virtue than a drawback. Using an easily applicable network and training strategy can ensure reproducibility by other researchers. Moreover, the level of generalizability and robustness implies that there is limited room for improvement in this type of classification and regression tasks. Considering the rapid changes in the neural network field, the demonstrated robustness and generalizability may be more valuable than a novel breakthrough in neural network model design with marginally improved performance.

My comments and suggestions primarily focus on the broad generalizability and robustness of the work, with the aim of benefiting researchers who require quantification of protein aggregation using widefield images of model cell systems. The authors need not address all of my comments literally; if the study is actually aimed at more specific protein aggregations and readers, it would be sufficient to provide a detailed description of the network's applicability and limitations for that purpose.

We thank the reviewer for highlighting the robustness and generalizability of our work. Indeed, our aim was to use standard and widely-applicable neural network architectures and techniques to achieve our goal of label-free identification of protein aggregates, without sacrificing model accuracy. We managed to do this even with a U-Net architecture that has reduced model complexity (4x fewer feature maps), enabling high reproducibility of our work even when computational resources are scarce. We highly appreciate the reviewer's suggestions to enhance the generalizability and robustness of our work even further, and we have addressed their specific comments below.

Comments

1. Regarding model validation (Fig. 2c): The pixel regression network employed SSIM and Pearson correlation, but these two metrics may not be suitable for validation in this study. Pearson correlation generally underperforms in image comparison tasks, while SSIM aims to mimic human perception in image comparison. Consequently, when most of the output images consist of noise or background, and only a small ROI contains the actual signal (as is the case in this study), these metrics may produce large discrepancies in the results. Therefore, I suggest the following:

a. Present normalized MSE: Since MSE was utilized as the training loss, it would be beneficial to display the results in a human-comprehensible format for validation purposes.

We used SSIM and Pearson correlation for validation because we found these techniques to be standardized in the field of artificially-intelligent microscopy¹, for example, two of the seminal works showcasing virtual labeling^{2,3} use Pearson correlation as their validation metric. Additionally, they range between 0 and 1, easing their interpretation.

We agree that normalized MSE would also be useful since indeed MSE was used as the training loss. We have added the model's normalized MSE (NMSE) performance in Supplementary Fig. 3b (shown below), which shows agreement with the other two metrics. We also now only report the Pearson correlation coefficient in Fig. 2c for visual clarity and we

have removed SSIM from the entire paper, replacing it with the Pearson correlation and the NMSE.

Supplementary Figure 3: Quantitative validation of LINA. (a) Pearson correlation coefficient (r) computed only on the regions where there are aggregates. The metric is computed for the 8-plane-QPI pixel-regression model. (b) Normalized mean squared error computed only on the regions where there are aggregates. The metric is computed for the 8-plane-QPI pixel-regression model. (c) Jaccard index computed for the 8-plane-QPI pixel-classification model. (d) Dice coefficient computed for the 8-plane-QPI pixel-classification model.

b. Employ metrics with better pixel-wise accuracy representation: Generally, Dice loss and its variants are more suitable for this purpose. However, the authors can explore alternative metrics with appropriate justifications. If the authors prefer to maintain SSIM and Pearson correlation, it is advisable to crop images around the ROI or apply a certain threshold. The specific approach for generating these metrics should be described in the methodology. The same cropping or thresholding technique can also be applied when representing the MSE numbers (Comment 1-a) since the background MSE might be somewhat significant due to the large area it occupies.

We agree that Dice loss and its variants are very useful metrics for quantifying pixel-classification performance. In our original manuscript, we had included one of the variants of the Dice loss, the Jaccard index (also referred to as Intersection over Union/IOU). We have now calculated it for our pixel-regression model as well (after segmenting the predictions using Otsu thresholding), and we now also calculate the Dice loss. Both metrics are now shown in Supplementary Fig. 3c, d (shown above).

We believe that quantifying the pixel-regression metrics over the entire image/field of view is crucial to also quantify the amount of false positives/true negatives, and not just true positives and false negatives (which can be quantified if we crop the images around the true aggregates).

We, therefore, respectfully disagree that most of the output images should be considered noise or background. The field of view regularly includes cells that do not have aggregates. It is crucial to determine, in these regions and even for other cellular regions within a cell containing an aggregate, whether the model can accurately identify the absence of aggregates (true negatives), as well as whether the model hallucinates and falsely predicts an aggregate (false positives).

However, for more precise quantification of just the true positives and false negatives, we have now added the quantitative performance of the two metrics (Pearson correlation and NMSE) on cropped ROIs (aggregates) in Supplementary Fig. 3a, b (shown above). We still observe excellent quantitative performance (mean $r = 0.955$ and mean NMSE = 0.106). We describe the methodology for cropping the ROIs and calculating the metrics in the Methods section (line 639).

Below is the discussion of these metrics that is now written in the revised manuscript (line 198):

The correlation coefficient is consistently high (0.959 mean) over the entire test set, a set of images which the model had never seen before, illustrating the high reliability of the model. We further evaluated the quantitative performance of the model by measuring r and the normalized mean squared error (NMSE), the quantity used for training (unnormalized), computed only within the regions where there are aggregates (Supplementary Fig. 3a, b), thereby more precisely measuring the degree of true positives/false negatives (Methods). The correlation remains very high (0.955 mean), and the NMSE also consistently shows relatively low errors (0.106 mean). The pixel-classification model also performs well, achieving an F1-score of 0.9 and a mean Jaccard index of 0.78. The pixel-regression model is slightly more accurate when segmentations are produced from its regression predictions, with a mean Jaccard index of 0.81 and a mean Dice loss of 0.89 (Supplementary Fig. 3c, d). Therefore, we focus on the regression model for our following experiments and quantifications.

c. Complement metric representation with intensity comparisons: In addition to comparing the pixel-wise similarity of ground-truth and network-output images using metrics, it is essential to compare the total intensity of ground-truth fluorescence images and network-output images for aggregations. Without this validation, the network cannot be effectively used for quantification purposes in kinetics studies, as its dry mass cannot be quantified without supplementary dry mass information (e.g., QPI-based dry mass images in this study).

We thank the reviewer for raising this point. We now include total intensity comparisons in line 210, as follows:

We used the Pearson correlation coefficient to compare the total intensity within the aggregates in the prediction images versus the ground truth, which was computed as 0.91, suggesting very high correlation in this regard as well.

This is calculated using the cropped ROIs described in our response to Comment 1b.

2. Regarding Generalizability: Although the training and validation of the network are based on Httex1 aggregates and their variants, the network's robustness and consistent performance across different Httex1 variants suggest that it may be applicable to various other protein aggregates in general. This implies that the network might have learned features unique to aggregates, distinguishing them from cellular component features. It appears that the authors intended to emphasize this aspect, as evidenced by the manuscript's title. However, there are a few concerns to address:

a. Circularity: The images presented in the main figures and the distribution of morphology (Supplementary Fig. 6) show that aggregates possess highly circular morphologies. However, fibrillar morphology is frequently observed in NDD studies [1]. Moreover, more diverse morphologies can arise when considering that aggregates can form alongside interacting cellular organelles [2]. It would be beneficial for the authors to address this issue. One possible approach is simulating non-circular morphologies in their test/validation images. Given that the authors already have a well-performing classification network, they could use it to determine the aggregate's boundary and introduce distortions in the subregions to generate non-circular aggregation images. Alternatively, the authors could add a note explaining why circularity does not impact generalizability.

[1] Boatz, J. C., Piretra, T., Lasorsa, A., Matlahov, I., Conway, J. F., van der Wel, P. C. A. (2020). Protofilament Structure and Supramolecular Polymorphism of Aggregated Mutant Huntingtin Exon 1. *Journal of Molecular Biology*

[2] Zhou, Y., Peskett, T. R., Landles, C., Warner, J. B., Sathasivam, K., Smith, E. J., Chen, S., Wetzel, R., Lashuel, H. A., Bates, G. P., Saibil, H. R. (2021). Correlative light and electron

microscopy suggests that mutant huntingtin dysregulates the endolysosomal pathway in presymptomatic Huntington's disease. *Acta Neuropathologica Communications*

We thank the reviewer for emphasizing the high robustness of our technique and its potential to be generalized to different kinds of protein aggregates.

We would like to point out that the work by Boatz and colleagues is done on in vitro fibrils, which is why they show the fibrillar morphology. At the ultrastructural level, beyond the diffraction limit and on a smaller scale, these aggregates are composed of filamentous amyloid fibrils of sizes 10-12 nm. However, on a larger scale, when using a diffraction-limited microscope, most cellular models of proteins involved in neurodegenerative diseases form aggregates that are spherical/circular-like, particularly for late-stage inclusions. Studies from our labs and others have shown that fibrillar aggregates indeed form, however, these aggregates generally transform into circular-like inclusions during the late stages⁴. Although circular-like inclusions are the dominant form of inclusions in diseased brains, we do acknowledge the polymorphism of protein aggregates and the eventual need for better methods to capture this diversity. We believe that this is beyond the scope of this manuscript and that our work will stimulate greater interest in developing more refined methods and approaches to achieve this goal.

Nonetheless, we agree that it is still interesting to quantify the performance of our model on less-circular/chimeric aggregates. Therefore, we generated a dataset of noncircular aggregates by masking parts of each aggregate with cellular images (a suggestion made in the next comment by the reviewer). We summarize the performance in Supplementary Fig. 6 (shown below) and discuss it in line 291 of the manuscript, copied below. The model generally performs very well, showcasing the high potential to apply LINA on different kinds of aggregates in other diseases.

Supplementary Figure 6: Validation of LINA for simulated input images of chimeric, non-circular aggregates. (a) Example images visualizing the model's performance on a non-simulated, circular aggregate. (b) The model's performance on the same example after replacing part of the aggregate in the input image and the ground truth with part of a cell/background image (and its corresponding fluorescence signal) to create a simulated test example of a chimeric, non-circular aggregate. The model is able to accurately predict the presence of the heterogeneous aggregate. (c) Pearson correlation

coefficient (r) computed on the simulated test set, only on the regions where there are aggregates. The metric is computed for the 8-plane-QPI pixel-regression model. (d) Normalized mean squared error computed on the simulated test set, only on the regions where there are aggregates. The metric is computed for the 8-plane-QPI pixel-regression model. Scale bars: 5 μm .

To test the potential of generalizing LINA to various kinds of protein aggregates involved in other NDDs, which could vary in terms of circularity and heterogeneity, we simulated a non-circular, chimeric test set and tested LINA's performance on it. This was done by replacing parts of aggregates in the test set by images of cells/background (for both the input and label), and then testing the ability of the pixel-regression model to correctly identify the heterogeneous aggregates (Supplementary Fig. 6). The Pearson correlation coefficient (mean 0.89) and NMSE (mean 0.08) show excellent performance even with these perturbations, demonstrating the high potential of generalizing LINA to different kinds of protein aggregates, of varying morphologies and structure.

b. Optical properties of aggregation: **The authors have demonstrated that the trained network is robust across different variants of Httex1 aggregations, and even outputs from brightfield images are quite reliable.** When considering protein aggregates from other proteins, differences will be manifested in their refractive indices, leading to phase differences in QPI or intensity variations in brightfield images. Consequently, the contrast between protein aggregation and cellular background images becomes critical. If the authors can present a working range of SNR concerning protein aggregation (signal) to cellular background (noise) for its robustness, it could justify the network's general applicability without conducting additional experiments on diverse protein aggregations. One approach could be to add noise specific to the protein aggregation region or blur the protein aggregation region with cellular images.

We agree that it is important to study the network's performance as the SNR of the input images is varied. We had studied this in two ways: artificially, by adding varying amounts of noise to our original test set, and experimentally, by testing the performance on a new dataset collected at shorter exposure times.

We now extend the first noise study, as the reviewer suggested, by only adding noise to our test set in the regions where we have aggregates. This is now presented in a revised Supplementary Fig. 5 (shown below).

Supplementary Figure 5: Robustness of the trained LINA models. (a) The pixel classification network can outperform the label and correctly identify unsegmented aggregates. A test set example is shown in which an aggregate was correctly identified by the neural network, despite being missing in the segmented ground truth label due to a segmentation error. The fluorescence ground truth signal confirms the presence of an aggregate in those pixels. (b), (c), (e), (f), and (g) show the pixel-regression model's behavior when noise is artificially added. (b) The model is still able to identify the aggregate in a test set example in which a small amount of Gaussian noise is added. (c) Images of a test set example at an extreme level of artificially added noise (0.48 rad^2 variance). The model is still able to identify the aggregate after the input is pre-filtered, despite both the original and pre-filtered phase images looking like pure noise. (f) At larger amounts of artificially added noise, the model performs worse quantitatively, however, this is mitigated by pre-filtering (blurring) the images prior to them being given as input to the network. With pre-filtering, the model remains stable even at extremely large (unrealistic) amounts of noise, as shown in (c). (e) Images of a test set example where noise (0.2 rad^2 variance) is artificially added to the region where there is an aggregate. The model is still able to identify the aggregate after the input is pre-filtered. (g) Similar to (f), the quantitative performance is maintained after pre-filtering but degrades slowly until it saturates. (d) and (h) illustrate LINA's robustness at experimentally-varied signal-to-noise ratios. (d) With an image acquired at a 3 ms exposure time, LINA is still able to identify aggregates, even though it was trained exclusively using images taken at a 50ms exposure time. (h) Summary of LINA's quantitative performance for input images taken at different exposure times, showing high accuracy and consistency. Error bars represent the standard deviation. Scale bars: $5 \mu\text{m}$.

Below is a discussion we added in the manuscript regarding this point (line 260):

By adding noise specifically to the aggregate regions and with the same pre-filtering step, we show that LINA has high potential to be applied on aggregates of varying optical properties. We observe, both visually and quantitatively, that the performance degrades when noise is added that is larger than or equal to 0.5 rad^2 , so we consider this the acceptable range for our model, though it can be extended in the future through transfer learning.

As mentioned in the previous comment's response, we have also verified the network's functionality when we blurred the aggregate regions with images of cells (Supplementary Fig. 6).

c. Sharing the training weights (i.e., the pre-trained model) is highly recommended. The generalizability claims are based on the trained network, rather than the network architecture or training strategy. In my opinion, if the trained model is not publicly available, the generalizability loses much of its value. However, I could not find any mention of the availability of the pre-trained model in the current version of the manuscript.

We thank the reviewer for this important comment. It has always been our plan to make the trained model available for the community in the most easily-applicable way. We had provided the pre-trained model to the editor during the submission process to be made available for reviewers, however, we were still working on annotating the code to ensure that we can present it in the most user-friendly way.

We now have made the trained model and code available on GitHub, under the following link: <https://github.com/kibb/LINA/>

We describe this in line 676.

d. Specification of the detection limit: Supplementary Fig. 6 reveals that the training data do not include small-sized aggregates. It would be beneficial to understand the extent to which the network can identify and quantify protein aggregations for use in kinetic studies. Additionally, it would be helpful to provide information on false positives, such as presenting representative outputs for cell images without aggregation as examples.

The training data includes aggregates with areas as small as 1.5-2 μm^2 and can detect aggregates as small as 3 μm^2 . This detection limit was mentioned in lines 116 and 202 of the originally-submitted manuscript.

The detection limit largely depends on the training dataset. This includes the nature of the data (e.g., type of aggregates that are acquired), the sample properties (refractive index variation, etc.) and optical limitations of the setup used for data acquisition (contrast, SNR, the diffraction limit, etc.). For our training dataset, we focused on late-stage aggregates. However, in future work, LINA can be extended and trained on intermediate-stage and early-stage aggregates. Indeed, we have already started working towards this goal. Collecting a large dataset of aggregates as they are forming, at this magnification and field of view, is challenging because we are limited in the region that we are looking at, not all cells will produce aggregates, and the process is rapid and could occur at any time after transfection up to ~48 hours. Once we have collected enough data, we aim in future work to extend the network to predict even smaller aggregates than what is currently possible.

We have added a paragraph discussing this in line 544 (shown below)

The detection limit of LINA depends on several factors, notably the size distribution of the training dataset and the limitations in the images resulting from the microscope (contrast, SNR, resolution) and the sample (transparency, refractive index heterogeneity). In our training dataset, we focused on late-stage aggregates, making sure to include images of aggregates of varying morphologies and sizes, such that the network can be as general as possible. We show that LINA works well for varying image SNRs, aggregate sizes and shapes, protein constructs, cell lines, and in live-cell imaging conditions. The current version of LINA can detect aggregates with areas as small as 3 μm^2 . To further enhance LINA's capabilities for kinetics studies, we aim in future work to expand the training dataset to include early and intermediate stage aggregates, which would enable the detection of even smaller aggregates than what is currently possible. This can be accomplished by re-training a new model on the expanded dataset, or through transfer learning on a dataset consisting of the earlier-stage aggregates.

Reviewer #3 also suggests (comment 2, page 17 of this document) that we try imaging at a smaller magnification, which is a very interesting idea and could in theory ease the collection of aggregate formation data. We have shown in response to Reviewer #3 that LINA works on a commercial setup with lower magnification (20X) (Supplementary Fig. 9). However, this approach would likely run into limitations of resolution. With a lower numerical aperture, the resolution will suffer, so it will likely not be possible to detect small aggregates. The diffraction limit, the properties of the sample such as transparency and refractive index, and the contrast/SNR of the images are what will ultimately limit what can possibly be detected from the input brightfield/quantitative phase images. We have already characterized LINA's performance for various imaging conditions, including a varying SNR, as discussed in an earlier comment.

We agree that it is helpful to show the network's performance in terms of false positives, which is why we had performed our validation and quantified our metrics on the full images, as described previously. We now also provide data of the model's performance on two different negative controls, cells expressing 16Q-GFP (which does not produce aggregates) and eGFP alone (Supplementary Fig. 4, shown below), where the model is able to correctly predict the absence of aggregates.

Supplementary Figure 4: Validation of LINA using negative controls. (a) Example images visualizing the pixel-regression model's performance on a negative control, 16Q-GFP. Protein aggregates are not formed at this polyQ length. (b) Example images visualizing the pixel-regression model's performance on a negative control, eGFP, where there is no Httex1 protein at all. In both (a) and (b) the model successfully predicts the absence of aggregates. Scale bar: 5 μm .

And we have added the following summary in line 209:

Supplementary Fig. 4 shows LINA's performance on two negative controls (cells expressing 16Q-GFP and eGFP, respectively), where aggregates are not produced. LINA successfully predicts the absence of aggregates in both cases.

We are not able to compute the Pearson correlation coefficient or the NMSE because the ground truth is a zero-image. However, we provide below the SSIM to illustrate the consistently-accurate predictions:

Revision Figure 1: Quantitative validation of LINA on negative controls. (a) SSIM computed on images of cells expressing 16Q-GFP (mean 0.85). (b) SSIM computed on images of cells expressing eGFP (mean 0.89).

The following suggestions may not be critical for this study and could be beyond its scope. However, in my humble opinion, these recommendations could enhance the study's value for the scientific community and related research fields.

We thank the reviewer for the valuable and constructive feedback and recommendations aimed at further improving the significance and impact of our work.

Suggestions

1. Since QPI might not be readily available in many biology labs, it could be interesting to examine the network's performance on phase-contrast and DIC images, which are more commonly accessible. Furthermore, if the network has indeed learned to distinguish protein aggregation features from cellular features, it is possible that it may also work on EM images without additional training.

We agree with the reviewer that QPI might not be as simply available in all biology labs, although with our previously-published simple phase reconstruction algorithm⁵, this could be

somewhat mitigated. We now also show results on QPI images produced from images acquired on a commercial, confocal setup (Supplementary Fig. 8 and 9), further showcasing the simplicity of using our phase-reconstruction algorithm. Additionally, we hope that by providing models that can work very well even with simple brightfield inputs (which are even more basic than phase contrast or DIC), this could make our approach even more accessible for labs around the world.

It is a very interesting suggestion to test or extend the approach to EM images, however, we believe this is outside the scope of the current study. We added the following in line 571 of the manuscript:

Additionally, LINA could potentially be extended in the future to virtually label aggregates from electron microscopy images.

2. An online server: Although machine learning and neural networks are becoming increasingly popular, many researchers still lack the resources and expertise to utilize published models effectively. To maximize the benefits for both authors and the scientific community, it would be advantageous to create an online server where users can simply submit an image and receive output from the pre-trained model. Given the generalizability and robustness of this network, it has the potential to become a standard label-free quantification method. By including a note encouraging users to cite the relevant paper when using this resource, the authors could generate significant impact and recognition for their work. Imagine the number of citations that could be received if every time someone used SDS-PAGE products or employed the theory of relativity in their research, they were asked to cite the original electrophoresis or the relativity theory paper. Unfortunately, Einstein and the developers of electrophoresis forgot that notes, even though they received Nobel prizes. But you can put that note! Though a Nobel prize is not guaranteed.

We thank the reviewer for this very interesting suggestion. Indeed, we plan to do our best to make the models as easily available and usable for researchers. Unfortunately, it is currently not feasible for us to run an online GPU server for this task. However, we hope that our annotated, open-source code, provided on GitHub, can still be easily adopted by scientists. We believe that tools such as ImJoy⁶, ZeroCostDL4Mic⁷, or DeepImageJ⁸, can also be very useful in easing the use of our deep learning model.

3. Heterogeneity. This point is an extension of Comment 2-b, but addressing heterogeneity could be a significant undertaking, so it is presented as a separate suggestion. Structural variance, particularly when it involves structural heterogeneity, can result in inhomogeneous optical properties [3]. Therefore, it would be worthwhile to explore whether the network performs well with these heterogeneous aggregations. Once again, this could potentially be achieved by creating chimeric aggregation images from the training images, allowing for a more comprehensive assessment of the network's performance.

[3] Meng, F., Yoo, J., Sung, H. (2022). Single-molecule fluorescence imaging and deep learning reveal highly heterogeneous aggregation of amyloid- β 42. Proceedings of the National Academy of Sciences

Indeed, this is a very interesting suggestion. We have now tested the network's performance on chimeric aggregate-cell images (Supplementary Fig. 6), which showed that the model performs well even on heterogeneous aggregates with inhomogeneous optical properties.

In conclusion, I believe the paper is nearly complete as it stands, particularly in terms of the experimental validation of the trained network. The authors have demonstrated careful and thorough addressing of every aspect to ensure the network's generalizability within the current scope of the study. However, I believe this work has the potential for even broader generalizability. If this appeals to the authors, I hope my other

comments will be helpful in revising the manuscript accordingly or future follow-up studies. Please note that not all of my comments are of critical importance (except for 1-a and 1-b), and they should be used as guidance in line with the authors' revision direction. These points will be reiterated to the editor.

We are very grateful to the reviewer for their insightful and valuable comments, and we would like to thank the reviewer for the positive assessment of our manuscript and for appreciating the high generalizability of our work.

Reviewer #2:

Ibrahim and coworkers develop a CNN based approach to identify and quantify aggregates in bright field images of Huntington's disease cellular models. The authors perform appropriate training and validation of LINA (the proposed tool) demonstrating that it performs well on unseen samples. The idea behind the approach is to enable fast, accurate, broadly-applicable ultra-structural characterisation of aggregates implicated in neurodegenerative diseases. To address this mission, I have a number of important concerns that need to be addressed before I can recommend this for publication:

1, in their literature review, the authors do not mention an important class of in situ fluorescent reporters that are minimally invasive, broadly applicable to small and large aggregates of different proteins from different samples, and bright enough to enable to super resolution imaging. These reporters range from epitope and conformation specific antibodies / nanobodies to conformation specific, low Kd binders (such as ThX, YOYO, etc.). Why is a label free approach, that is less accurate in its quantification than a much more broadly applicable fluorescence based approach and as quick, be favoured?

We thank the reviewer for allowing us to address this point. As mentioned in the text, several studies, including recent work from our lab and others using advanced correlative electron microscopy, proteomics, and functional studies, have shown that the addition of non-native sequences such as fluorescent tags (e.g. GFP) or short peptide tags (poly-Histidine or HA tag) not only alter the ultrastructure properties and biochemical composition of Htt inclusions, but also their toxicity and interaction with other cellular components⁹⁻¹². Therefore, while tagging fluorescent proteins to Htt and other amyloid-forming proteins does not seem to significantly alter their ability to form fibrils, they do interfere with the formation of bona fide inclusions that resemble those found in patient brains. Thus, such models are not suitable for investigating Htt inclusion formation, maturation, and clearance.

Use of conformational antibodies;

Firstly, we are not aware of any robust and validated antibodies that are specific to Htt inclusions and do not recognize soluble Htt. Secondly, the use of Htt-specific antibodies does not allow for monitoring the dynamics of Htt aggregation, inclusion formation, and maturation and may interfere with the dynamics of Htt inclusion interaction with other proteins or cellular organelles.

Several studies have reported inconsistencies in the use of Thioflavin T/S to image amyloid proteins and inclusions in cells. While ThT/S is often useful to detect amyloid aggregates in cells and brain tissues, for many amyloid proteins, including aSyn and Htt, they give inconsistent results and are thus rarely used as standard tools to quantify pathology. Furthermore, several studies have reported that ThT/S do not capture the diversity of amyloid fibrils and some types of aSyn and Htt aggregates are not detected by ThT/S¹³. These dyes bind only to the fibrillar form of amyloid proteins and thus are not suitable for investigating the early stages of protein aggregation and inclusion formation. Finally, being amyloid-specific, they do not capture changes in the biochemical complexity and physical properties of the inclusions.

With respect to the use of YOYO as an amyloid dye, we could only identify one paper describing the use of this molecule to detect Ab42 amyloid fibrils:

<https://pubmed.ncbi.nlm.nih.gov/26612254/>

We could not find any other papers on the use of this dye to detect and image amyloid proteins or inclusions in cells or brain tissues.

2, aggregates, at least from other proteins such as tau, aSyn, and abeta, exhibit different conformations in different diseases, under different recombination conditions, and between different samples. Given that cellular models do not recapitulate human disease pathology, how well is the proposed approach suited to characterise aggregates from disease relevant samples (bio fluids, post mortem brains, and human-derived organoids) and different proteins and why has this not been shown.

First, we would like to point out that the primary objective of our work is to develop methods to detect and monitor the dynamics of amyloid inclusion in cells and not in cell-free systems. This is because it is now clear that pathological inclusions are composed of not only amyloid fibrils but a complex mixture of amyloid fibrils, lipids, membranous organelles, proteins, and other co-factors. Our ultimate goal is to develop methods that are insensitive to differences in the conformational properties within the inclusions. We view this as the main advantage of label-free methods in comparison to epitope or conformation-specific tools which are unlikely to capture the diversity of inclusions. The work we present here represents the first proof-of-concept study using label-free methods to monitor the dynamics and kinetics of inclusion formation in cells. We believe that this represents an important advance that will stimulate more interest in advancing this method and its application to investigate pathological inclusions in cellular and animal models as well as in post-mortem human brains.

Further studies are underway in our laboratory to extend its application to other types of inclusions, e.g. Lewy-body-like inclusions in neuronal models of PD. In particular, we are using cellular models that we have recently been developed and were shown to recapitulate Lewy body formation with a pathological diversity similar to that observed in PD brain^{4,14}.

We acknowledge that processes of protein aggregation and amyloid formation are influenced by the cellular milieu and amyloid proteins could form different types of aggregates/inclusions at different stages and different brain regions. Unfortunately, at this stage, there are no cellular models of Htt that could faithfully reproduce the diversity of Htt pathological diversity in the brain and we lack understanding of the biochemical and ultrastructural properties that distinguish one type of aggregates from others. Furthermore, the appropriate tools, antibodies or amyloid probes, that would enable selective detection and monitoring the formation of different types of aggregates do not exist.

We now include results which show the high accuracy of LINA on images of non-circular, heterogeneous aggregates (Supplementary Fig. 6, shown below). This result shows the potential of using and extending LINA to diverse types of protein aggregates. We generated this dataset of noncircular aggregates by masking parts of each aggregate with cellular images (a suggestion made by Reviewer #1). We discuss this result in line 291 of the manuscript, copied below.

Supplementary Figure 6: Validation of LINA for simulated input images of chimeric, non-circular aggregates. (a) Example images visualizing the model's performance on a non-simulated, circular aggregate. (b) The model's performance on the same example after replacing part of the aggregate in the input image and the ground truth with part of a cell/background image (and its corresponding fluorescence signal) to create a simulated test example of a chimeric, non-circular aggregate. The model is able to accurately predict the presence of the heterogeneous aggregate. (c) Pearson correlation coefficient (r) computed on the simulated test set, only on the regions where there are aggregates. The metric is computed for the 8-plane-QPI pixel-regression model. (d) Normalized mean squared error computed on the simulated test set, only on the regions where there are aggregates. The metric is computed for the 8-plane-QPI pixel-regression model. Scale bars: 5 μm .

To test the potential of generalizing LINA to various kinds of protein aggregates involved in other NDDs, which could vary in terms of circularity and heterogeneity, we simulated a non-circular, chimeric test set and tested LINA's performance on it. This was done by replacing part of each aggregate in the test set by images of cells/background (for both the input and label), and then testing the ability of the pixel-regression model to correctly identify the heterogeneous aggregates (Supplementary Fig. 5). The Pearson correlation coefficient (mean 0.89) and NMSE (mean 0.08) show excellent performance even with these perturbations, demonstrating the high potential of generalizing LINA to different kinds of protein aggregates, of varying morphologies and structure.

3, the aim behind the proposed approach is to enable better understanding of the role of aggregation in disease progression, however, the authors didn't demonstrate any new functional insights from the application of LINA to their cellular models beyond measurement of dry mass over time.

We apologize for failing to point out that the cellular model of Htt inclusion formation we used in this study has been extensively characterized at the ultrastructural, biochemical, and functional levels. A full manuscript⁹ was devoted to characterizing the biochemical and ultrastructural properties of mutant Htt aggregation and inclusion formation in the presence and absence of GFP and the interplay between Htt (tag-free and Htt-GFP with variable polyQ repeats) aggregation, inclusion formation, cytotoxicity, and organelle dysfunction in both

mammalian cells and neurons. In fact, it was the findings from this model that highlighted the need for label-free methods and inspired this work.

Our primary goal in developing LINA was to provide a label-free method for higher-fidelity, dynamic and rapid analysis of protein aggregation. We demonstrated its ability for real-time monitoring of aggregates as they form. We showed the robustness and generalizability of our method over various conditions, including imaging setups, protein constructs, and cell lines. In our current study, we demonstrated an increased mean area and dry mass for shorter-polyQ-length aggregates, which was similar to the GFP-labeled aggregates, and correlates with their different aggregation mechanisms. We primarily focused on the measurement of intensity, dry mass and area changes of Httex1 aggregates over time, and we acknowledge the importance of demonstrating additional functional insights.

We are actively working with collaborators on expanding the scope of LINA's applications to gain more comprehensive insights into the dynamics and behaviors of aggregates in cellular models. We believe our tool will enable researchers in the community to perform various other studies.

4, the authors use custom built instrumentation and a LabVIEW software for data acquisition automation. Can the workflow be demonstrated on a commercial microscope and an open source data acquisition tool to demonstrate general use. The same could be also though for the training model - which having looked at the having available source code - is not sufficiently annotated or packaged into a GUI-based software to be used by biologists who might be interested in applying the tool.

LINA is a post-processing algorithm. Therefore, in theory, it can be used with data acquired on any microscope (as long as the images have sufficient SNR/information), and the software used for data acquisition does not influence the quality of the predictions. However, we agree that it is interesting to study LINA's performance on different imaging setups to show the generalizability of the method. We had previously tested it on a different setup than our standard one (where we did the training and validation) and showed that it is possible to identify aggregates even from 1-plane brightfield images on that setup.

We have now extended our study and, as the reviewer suggests, we demonstrate LINA's excellent performance on a commercial microscope (a Leica TCS SP8 confocal microscope) in Supplementary Fig. 8 (shown below).

Supplementary Figure 8: LINA accurately predicts aggregates imaged on a commercial microscope. (a) Example images showing LINA's performance on an 8-plane QPI input. (b) Example images showing LINA's performance on a 1-plane brightfield input. Scale bars: 5 μm .

We acquired 8-plane images on this microscope and extracted the phase information using our QPI algorithm⁵. We show that the network accurately identifies aggregates with this kind of input (Supplementary Fig. 8a) and even with 1-plane brightfield inputs (Supplementary Fig. 8b). Even though this microscope is confocal and not widefield, has different illumination, detectors, and objectives, the model is still able to accurately identify the aggregate without needing transfer learning, showing the high generalizability and ease of use of our method.

This result (as well as the result described in reply to Reviewer #3's comment 2 in page 17 of this document) are also a demonstration of the ease of use and the accuracy of using our QPI algorithm on a setup that is different from our own.

We have added the following paragraph summarizing this result in line 333:

Next, we tested LINA on a commercial, confocal setup (Leica TCS SP8). We used a similar magnification (63X versus 60X) and NA (1.4 versus 1.2) objective, but the illumination, setup type (confocal versus widefield) and detectors were completely different. Here, we acquired 8-plane stacks and used our QPI algorithm⁵ to produce 8-plane QPI stacks. We tested LINA's performance using these QPI inputs (Supplementary Fig. 8a), as well as using the simplest possible inputs, 1-plane brightfield (Supplementary Fig. 8b), and both types of inputs yielded highly-accurate network predictions.

Additionally, we have made our code and models available on GitHub, under the following link:

<https://github.com/kibb/LINA/>
and we describe this in line 676.

We added more documentation in the repository for how to use the code, as well as annotations in the code. We believe that it will be fairly simple to use for biologists, as we provide a Jupyter notebook for each code, where all that is needed is to set the correct paths or stick to the default paths defined in the code.

Reviewer #3:

In their paper “Label-free Identification of Protein Aggregates Using Deep Learning” by Ibrahim et al., the authors present an approach (LINA) to study protein aggregation without using fluorescence labeling other than during the training of a machine learning algorithm. The authors are using quantitative phase reconstruction from multiple plane brightfield imaging to obtain both virtual staining used for segmentation of aggregate and dry mass quantification of these lasts. The authors claim the generalizability of their machine learning approach over different variable such as the type of protein, cell type, live/fixed, type of imaging (BF of phase), microscope variation and noise.

The overall paper is well written and organized with convincing results which are mainly impacting the biological community and rely on well-known experimental tools and algorithm strategies. I suggest some improvement on the paper to make it more attractive and improve the spread in a larger community.

We thank the reviewer for their positive comments about our work and we appreciate their suggestions to enhance the applicability of the work to a larger community, which we have addressed below.

- My major concern is regarding the ambiguity of fluorescence labeling. Indeed, as very clearly stated in the paper, the use of fluorescence labelling may perturbate aggregation. However, to train the algorithm, fluorescence labelling has to be performed. In the study case proposed by the authors, the cell line and labelling strategy are chosen not to perturbate the aggregation but how to generalize this to other protein aggregations since live imaging for dynamic aggregation/de-aggregation study is researched? Does immunohistochemistry perform as good as GFP based transfection for the training since it might be the general procedure? This is demonstrated with a training on transfected cells applied on histochemistry stained cells but does the opposite still work?

This is a very good point, as indeed it is often necessary to use immunohistochemistry and immunocytochemistry in other models of neurodegenerative diseases. It has been shown in prior studies^{3,15} in the field of virtual labeling that it is indeed possible to train successful models for predicting the fluorescence signal of cellular constituents, including aggregates, using labels obtained from immunocytochemistry.

Generally, any label should work for training purposes as long as it accurately signals the pattern that the neural network should learn and can be distinguished from the background. This can even be manual annotations, which is often done in the field of machine learning, assuming the annotator has a way of identifying which parts of the image display the effect that they would like the network to learn to identify. Often, this is the case when human visual inspection is enough for accurate identification, such as labeling pictures of dogs or cats, or even for more specialized tasks, such as when physicians label the presence or absence of tumors.

We have shown in our work (Supplementary Fig. 5a in the revised manuscript) that even if there are some errors in the training labels, the network can nonetheless learn the correct relationship and neglect these errors. However, care should be taken, as this is not always the case, and incorrect or unnecessary information can lead to hallucinations¹⁶.

- My second concern is on the observation scale. Indeed, clusters are large (>3 μ m) and the numerical aperture and magnification used are important. Is it a requirement (for accurate phase reconstruction for ex.)? Indeed, the authors are using a automatized XY-scan to have large FoV but it would be more interesting to have the same study directly with low mag and

NA objectives. I would recommend a test at low magnification since it could greatly enhance the spread of this technique with a huge enhancement in the acquisition speed.

We thank the reviewer for this interesting suggestion, which aims to make it easier for researchers to have a higher throughput when using our tool. As mentioned in our response to Reviewer #1 (comment 2d, page 7 of this document), our detection limit is roughly an area of $\sim 3 \mu\text{m}^2$, which corresponds to diameters slightly $< 2 \mu\text{m}$. Indeed, the numerical aperture, and hence the resolution, directly affect the accuracy of the phase reconstruction.

We have now performed experiments using a lower-magnification objective (20x versus 60x originally) and NA (0.85 versus 1.2 originally) on a commercial setup. We acquired 8-z-plane image stacks and used our QPI algorithm⁵ to produce 8-plane QPI stacks, to be used as input to our network. Since the network expects 352 pixels x 352 pixels inputs, we first cropped regions of that size around the aggregate images and used those as inputs to the network. We also tried creating a model instance which expects an input size of 1024 pixels x 1024 pixels and giving the 1024 pixels x 1024 pixels images directly. In both cases, the network performs very well and accurately identifies the aggregates. We expected to possibly need transfer learning so that the network learns to expect smaller features (due to the lower magnification), yet the model works very well directly, illustrating the high generalizability and robustness of our method. This result (as well as the result described in reply to Reviewer #2's comment 4 in page 15 of this document) are also a demonstration of the ease of use and the accuracy of using our QPI algorithm on a setup that is different from our own.

We show this result in Supplementary Fig. 9 (shown below) and we have added the following paragraph summarizing the result in line 343:

Supplementary Figure 9: LINA accurately predicts aggregates imaged using a lower magnification (20X) and NA (0.85) objective. (a) Example images showing LINA's performance on an 8-plane QPI input, which is cropped around the aggregate to have the standard input size of 352 pixels x 352 pixels. Scale bar: 5 μm . (b) Example images showing LINA's performance on the 1024 pixels x 1024 pixels input directly. Scale bar: 20 μm .

We further tested LINA on the same commercial setup, with a lower-magnification (20X versus 60X) and NA (0.85 versus 1.2) objective (Supplementary Fig. 9). A lower-magnification objective means the FOV will be larger and, therefore, would enable users of our method to have a higher imaging throughput of cells containing aggregates. Since the network expects

352 pixels x 352 pixels inputs, we first cropped regions of that size around the aggregate images and used those as inputs to the network. We also directly tested the model on the 1024 pixels x 1024 pixels images. In both cases, the network performs very well and accurately identifies the aggregates. Here, again, we expected to possibly need transfer learning so that the network learns to expect smaller features (due to the lower magnification), yet the model works very well directly, illustrating the high generalizability and robustness of our method. These results using QPI inputs generated from images acquired on the commercial setup also demonstrate the ease of use and the accuracy of using our QPI algorithm on a setup that is different from our own.

We agree with the reviewer that a lower magnification increases the throughput and reduces the imaging time. However, a caveat is that at a lower magnification/numerical aperture/resolution, there will likely not be enough information to identify aggregates beyond a certain size, due to the diffraction limit and constraints of transparency and contrast. This is particularly a problem when using our tool for live imaging, as aggregates will start out quite small. It is already quite difficult to identify aggregates at this level of magnification and resolution from label-free images. This is evident from a previous study¹⁵, which had to perform expansion microscopy to be able to train a neural network to identify fixed, late-stage aggregates, while we were able to identify large and small aggregates in live, unperturbed samples.

Therefore, we have added the following paragraph to discuss this caveat in the manuscript discussion in line 560:

Methods to increase the throughput of the data acquisition, such as the automatic XY-scan mode we use here, are useful for collecting new data for transfer learning or re-training. Another possibility would be to image at a lower magnification, increasing the yield from one image, which we showed is a possibility using our method (Supplementary Fig. 9). However, this comes at a cost of resolution and information, which will likely complicate the identification of small aggregates from transmitted-light images, which could be particularly problematic for live-cell imaging of aggregates as they form.

- The use of phase rather than just intensity is giving a very little gain, this has to be stated in the abstract by removing the requirement of using phase information until the discussion on the dry mass. This will strengthen the generalizability of the LINA approach.

We have modified the abstract accordingly. We now use the term 'transmitted-light images' rather than 'quantitative phase images'.

- Since the reconstruction is 3D for both phase and fluorescence, why no 3D study has been performed? This has to be done or at least discussed in term of memory size, speed, gain in segmentation...

When designing the study, we aimed to use our multiplane microscope, knowing that this task is challenging and that the more information we give to the neural network, the more likely it is that it would be successfully trained. Additionally, a previous virtual-labeling study³ had shown the enhanced performance of the trained model with an increasing number of input planes (until saturation).

In this work, we achieve 2D identification of label-free aggregates. We train the neural network using 3D input images to maximize the amount of information given to the network, as mentioned above, and 2D 8-plane maximum projection fluorescence images as labels. This is for several reasons:

1. Memory constraints: 3D prediction would require 3D convolutions and 8x the number of labels. This would necessitate much higher computational resources (mainly GPU memory) and the method would, therefore, be more difficult to be accessible for the community.
2. Minimal variance of information in the 8-planes fluorescence images: we observed that there was minimal variance in the area/morphology of the fluorescence signal over the 8 planes, other than the signal going out-of-focus. This can be verified in the figure below:

Revision Figure 2: Example of 8-plane fluorescence images from the training dataset. There is no major variation in size/morphology.

However, there was more variation in the area of the aggregates in the QPI/brightfield images, which is why we believe it would be more accurate to identify an aggregate in 2D space and then infer its position within the other planes afterwards. Therefore, we instead aim to predict the maximum z-projection images, which combines information from the 8 planes, similar to previous virtual-labeling work³.

3. Lack of sufficient improvement: we did some initial tests to achieve 3D prediction and did not obtain promising results. The network was producing basically the same output for all 8 planes (likely due to point number 2 – that the difference in the labels is not large).

3D prediction is a challenging task as described in similar prior work³:

A note on 3D prediction

The approach we describe can in principle be applied to predicting 3D confocal voxel grids, and an early version of this work did incorporate a 3D prediction task with modest results.

There are at least three problems which must be overcome to make 3D prediction work:

1. Representation of the z-dimension in the network (low difficulty): In this paper, we simply merged the z and feature dimensions, which works when the number of possible z values is small but doesn't scale for a large number of possible z values. In that case, one would probably want to use 3D convolutions rather than 2D convolutions in the neural network.
2. Registration in z (higher difficulty): Independent pixel losses, such as the one we use, fail when input and target tensors are misaligned in an unpredictable manner. While we show it is possible to ensure registration in x and y across transmitted light and fluorescence images, we have not attempted to register in z.
3. Information (unknown difficulty): We suspect depth from blur in transmitted light will not be enough to recover 3D shape in multilayer cell cultures. It will take creative thinking to extract the information needed to reconstruct the 3D structure.

Revision Note 1: Some of the typical issues associated with 3D virtual labeling tasks. Issues 1 and 3 (also described in our listed issues number 1 and 3, respectively) particularly complicate this task in our case. Copied from work by Christiansen and colleagues³.

We have now added more information summarizing these points in line 163.

We aim to achieve 2D, rather than 3D, identification of the aggregates, for several reasons. A major reason is that this lowers the memory constraints considerably and enhances the accessibility of the method. 3D prediction is a challenging task as described in similar prior work¹⁷. Using our 2D-identification models, an aggregate can be identified in 2D space first and then its position within the other planes can be inferred from the QPI images afterwards.

We had also previously characterized (Supplementary Fig. 7) the variation in accuracy when training networks with fewer input planes (down to 1 plane, i.e. a 2D input). We show that LINA performs well even for 2D inputs, particularly for QPI inputs.

- I would like some details on the live study. The model used is trained on fixed cells. But what was the moment of aggregation (final I assume). Will it help to have a more diverse dataset with all stages of aggregation?

Our dataset consists of late-stage aggregates. However, we endeavored to include as diverse a dataset as possible, including aggregates with areas as small as $1.5 \mu\text{m}^2$, and with different morphologies.

The diversity of the training dataset determines the detection limit, which we discussed in previous comments from Reviewer #1 and Reviewer #3. It also determines the types of aggregates that can be recognized. If the training dataset is diverse enough, a general relationship can be learned by the network, and different kinds of aggregates (in terms of morphologies, underlying protein constructs, etc.) can be identified. We demonstrate in our results that our model works for various different conditions, as mentioned by the reviewer. This includes different types of protein, cell lines, SNRs, microscope set-ups, imaging modalities, numbers of input planes. We now have also shown the robustness of our model on chimeric, heterogeneous aggregates composed of part-aggregate and part-cell (Supplementary Fig. 6).

We are in the process of extending LINA with intermediate-stage and early-stage aggregates, which would enable even better kinetics studies, and we aim to report on this in future work. As mentioned in response to Reviewer #1 (comment 2d, page 8 of this document), achieving this goal requires the acquisition of a large dataset of high-temporal resolution aggregate formation events, which is challenging to capture at this magnification and field of view, as we are limited in the region that we are looking at, not all cells will produce aggregates, and the process is rapid and could occur at any time after transfection up to ~48 hours. Once we have collected enough data, we aim in future work to extend the network to predict even smaller aggregates than what is currently possible.

We have added a discussion about this in line 544 (shown below).

The detection limit of LINA depends on several factors, notably the size distribution of the training dataset and the limitations in the images resulting from the microscope (contrast, SNR, resolution) and the sample (transparency, refractive index heterogeneity). In our training dataset, we focused on late-stage aggregates, making sure to include images of aggregates of varying morphologies and sizes, such that the network can be as general as possible. We show that LINA works well for varying image SNRs, aggregate sizes and shapes, protein constructs, cell lines, and in live-cell imaging conditions. The current version of LINA can detect aggregates with areas as small as $3 \mu\text{m}^2$. To further enhance LINA's capabilities for kinetics studies, we aim in future work to expand the training dataset to include early and intermediate stage aggregates, which would enable the detection of even smaller aggregates than what is currently possible. This can be accomplished by re-training a new model on the expanded dataset, or through transfer learning on a dataset consisting of the earlier-stage aggregates.

- On the model transfer to other cell lines (ex. HeLa), I want to see images and not just SSIM curve for both the cells that are working well (high SSIM) and bad (lower SSIM). Does the virtual staining performance linked to a cell shape in particular? I also recommend to put it back in the main part of the paper the whole study since it is a major result for the algorithm generalizability.

We now show this result in the main results section (Fig. 3, shown below), rather than the SI, and have added representative images, the ones having the highest and lowest Pearson correlation.

Figure 3: Generalizability of LINA to a different cell line. (a) Pearson correlation coefficient (r) computed only on the regions where there are aggregates. The metric is computed for the 8-plane-QPI pixel-regression model. (b) Normalized mean squared error computed only on the regions where there are aggregates. The metric is computed for the 8-plane-QPI pixel-regression model. Both metrics show that LINA is able to accurately recognize aggregates expressed in a different cell line (HeLa). (c, d) Example images visualizing the model's performance on aggregates expressed in HeLa cells. (c) The example with the best r value is shown. (d) The example with the lowest r value is shown. Scale bars: 5 μm .

Even the lowest-correlation images are predicted very well, as can be observed. We believe the lower correlation could be due to having two aggregates simultaneously in the FOV (a fact that is well-predicted by the model), which might complicate predicting the intensity levels for both aggregates at the same time.

We have added the following discussion of this result in line 270:

Therefore, we quantified LINA's ability to identify Httex1-72Q-GFP aggregates expressed in HeLa cells, and the model consistently produced highly-accurate predictions, as quantified by measuring the Pearson correlation coefficient and the NMSE between the network outputs and the ground truth images (Fig. 3a, b). Fig. 3c, d show example images with the highest and lowest correlation coefficients, respectively. The lowest-correlation example still shows excellent performance, as both aggregates can be identified by the model. The reason for the slightly lower correlation could possibly be that having two aggregates in one FOV complicates the prediction of the true intensity levels for both aggregates at the same time, as we observed that the example with the highest NMSE (worst performance) also had two aggregates in the FOV, though this was also accurately predicted. These results indicate the high potential for LINA to be applied in other cell lines, greatly enhancing its generalizability.

- On the acquisition platform: I would like a SI image of the setup in order to have a complete overview of the method.

We have added a schematic of the setup in Supplementary Fig. 14 (shown below) and we refer to it in the Methods section.

Supplementary Figure 14: Schematic of the microscope setup used in this work. M – mirror, DM – dichroic mirror, MM – multi-mode, EM – emission. Adapted from work by Navikas and colleagues¹⁸. First developed and described in work by Descloux and colleagues⁵.

- On the data processing and AI: a more detailed section in the SI could greatly enhance the paper. With notably

- o a discussing on the presence or absence of overfitting and how it is checked.
- o a scheme of the U-net used.
- o a precision of the dimension of the input/output.
- o a global data processing pipe from the initial raw data down to the characterization of the clustering size and dry mass.

We had mentioned this information in the original manuscript in the Methods section.

We now also summarize this information in Supplementary Note 1 of the SI, copied below. We have also added a schematic of our neural network (Supplementary Fig. 15, shown below).

Supplementary Figure 15: Schematic of the neural network architecture used in this work (U-Net). Note that we also train models with fewer-plane inputs. Adapted from work by Ronneberger and colleagues¹⁹ and work from Ounkomol and colleagues².

Supplementary Note 1: Data processing pipeline and neural network training

We used custom MATLAB (R2021a) (Mathworks) scripts (available here) to retrieve the phase information from the brightfield images and produce quantitative phase images. These scripts are also used for pixel-registration in the 8 z-planes for both phase and fluorescence images. The images are cropped to a size of 352 pixels x 352 pixels.

We used Fiji (v2.9.0) scripts to produce maximum z-projection fluorescence images, to segment these images using Otsu thresholding and produce the labels for pixel classification, to prepare the color-coded maximum z-projection phase image shown in Fig. 2a, and for the image processing in Fig. 2b. Fiji was also used to segment network predictions and produce masks which are used to measure the area, circularity or dry mass of aggregates. Supplementary Fig. 13 summarizes the dry mass extraction process.

Our models are trained on a deep CNN with a U-Net¹⁹ architecture (Supplementary Fig. 15). Compared to the original architecture, we reduced the number of feature maps by a factor of 4 which led to a reduction in the trainable parameters by a factor of 15. This notably reduces GPU memory usage and training time, while still enabling excellent performance. We used TensorFlow (v2.8.0) and Keras (v2.8.0) to build our network, and training was done on a workstation equipped with an NVIDIA GeForce RTX 3090 GPU. The input images have dimensions of 352 pixels x 352 pixels in XY as aforementioned. Some models are trained on

8-plane inputs and some use fewer planes (4, 2, or 1). We used the adaptive moment estimation (Adam) optimizer with a learning rate of $1e-4$ and a mean squared error loss function. Before training our models, we normalize both the phase and fluorescence images by rescaling each image to be between 0 and 1. We used 10% of our dataset as the test set and split the rest of the dataset into training and validation sets with 20% being used for validation. We used the 'EarlyStopping' (on the validation loss) and 'ModelCheckpoint' callbacks to avoid overfitting and to save the best, most general models.

- I would prefer to see the discussion part on the minimal amount of data need to train LINA in the result part. It would fluidify the message.

We agree and have moved this section to line 356, as part of the Results section.

References

1. Laine, R. F., Arganda-Carreras, I., Henriques, R. & Jacquemet, G. Avoiding a replication crisis in deep-learning-based bioimage analysis. *Nat Methods* **18**, 1136–1144 (2021).
2. Ounkomol, C., Seshamani, S., Maleckar, M. M., Collman, F. & Johnson, G. R. Label-free prediction of three-dimensional fluorescence images from transmitted-light microscopy. *Nat Methods* **15**, 917–920 (2018).
3. Christiansen, E. M. *et al.* In Silico Labeling: Predicting Fluorescent Labels in Unlabeled Images. *Cell* **173**, 792-803.e19 (2018).
4. Mahul-Mellier, A.-L. *et al.* The process of Lewy body formation, rather than simply α -synuclein fibrillization, is one of the major drivers of neurodegeneration. *Proceedings of the National Academy of Sciences* **117**, 4971–4982 (2020).
5. Descloux, A. *et al.* Combined multi-plane phase retrieval and super-resolution optical fluctuation imaging for 4D cell microscopy. *Nature Photonics* **12**, 165-172, (2018).
6. Ouyang, W., Mueller, F., Hjelmare, M., Lundberg, E. & Zimmer, C. ImJoy: an open-source computational platform for the deep learning era. *Nat Methods* **16**, 1199–1200 (2019).
7. Chamier, L. *et al.* Democratising deep learning for microscopy with ZeroCostDL4Mic. *Nature Communications* **12**, 2276-2276, (2021).
8. Gómez-de-Mariscal, E. *et al.* DeepImageJ: A user-friendly plugin to run deep learning models in ImageJ. <http://biorxiv.org/lookup/doi/10.1101/799270> (2019) doi:10.1101/799270.
9. Riguet, N. *et al.* Nuclear and cytoplasmic huntingtin inclusions exhibit distinct biochemical composition, interactome and ultrastructural properties. *Nature communications* **12**, (2021).
10. Dahlgren, P. R. *et al.* Atomic force microscopy analysis of the Huntington protein nanofibril formation. (NBM, 2005).
11. Bäuerlein, F. J. *et al.* In situ architecture and cellular interactions of polyQ inclusions. *Cell* **171**, 179-187, (2017).
12. Shillcock, J. C., Hastings, J., Riguet, N. & Lashuel, H. A. Non-monotonic fibril surface occlusion by GFP tags from coarse-grained molecular simulations. *Computational and Structural Biotechnology Journal* **20**, 309-321, (2022).
13. De Giorgi, F. *et al.* Novel self-replicating α -synuclein polymorphs that escape ThT monitoring can spontaneously emerge and acutely spread in neurons. *Science Advances* **6**, eabc4364 (2020).
14. Kumar, S. T. *et al.* A NAC domain mutation (E83Q) unlocks the pathogenicity of human alpha-synuclein and recapitulates its pathological diversity. *Science Advances* **8**, eabn0044 (2022).
15. Lin, L.-E., Miao, K., Qian, C. & Wei, L. High spatial-resolution imaging of label-free in vivo protein aggregates by VISTA. *Analyst* **146**, 4135–4145 (2021).
16. Belthangady, C. & Royer, L. A. Applications, promises, and pitfalls of deep learning for fluorescence image reconstruction. *Nat Methods* **16**, 1215–1225 (2019).
17. Christiansen, E. M. *et al.* In silico labeling: predicting fluorescent labels in unlabeled images. *Cell* **173**, 792-803, (2018).
18. Navikas, V. *et al.* Correlative 3D microscopy of single cells using super-resolution and scanning ion-conductance microscopy. *Nat Commun* **12**, 4565 (2021).
19. Ronneberger, O., Fischer, P. & Brox, T. U-Net: convolutional networks for biomedical image segmentation. in *Medical Image Computing and Computer-assisted Intervention—MICCAI 2015* (2015).

REVIEWERS' COMMENTS

Reviewer #2 (Remarks to the Author):

I would like to thank the authors for their extensive response. I am convinced that the authors have made a commendable effort to open-source their codes and to clarify their rationale behind using the HTT cellular model in that study - and, therefore, happily recommend publication. However, as the authors prepare their final draft for publication I must indicate that I am not convinced of the authors' argument regarding the use of low affinity amyloid binders and that they are not capable of revealing the conformational diversity of aggregates / inclusions as much as the proposed label-free approach would do, and would like to ask the authors to consider / discuss the following in their final draft:

1, the authors are not the first to develop a label-free method to detect large aggregates within a cell-free system, please see:

<https://www.ncbi.nlm.nih.gov/pmc/articles/PMC6239428/>

and

<https://www.nature.com/articles/s42003-021-01981-x>

and

<https://iopscience.iop.org/article/10.1088/2040-8986/ac5b51/meta>

2, low affinity binders (and possibly a few high affinity binders such as antibodies) can target different conformations at the oligomeric, pre-fibrillar, and fibrillar levels of different proteins without altering (or substantially altering) the ultra-structure of the underlying aggregates allowing a very wide range of aggregates from small, nanoscopic oligomers to long, micro-scale sized fibrils to be imaged with high fidelity. A head to head comparison would show that such binders are easily capable of detecting a wider range of conformations across a longer length scale than the proposed label-free approach. It can be further argued that further use and development of the proposed approach will inevitably require the use of such binders to detect different conformations in a label-free manner.

3, mentioning that the proposed method is insensitive to the conformational properties within the inclusions is an unsupported statement, first because the imaged conformations are a few (the strong conformational diversity of misfolded protein aggregates is a genuine bottle neck in this research field), and second, the neural network is not sufficiently interrogated to know what kind of features it has learnt to make it capable of distinguishing one type of aggregates from the other and whether those features are extendible.

4, this work could have been done through fluorescent tagging, as the authors indeed do. And, generally speaking, the use of different imaging modalities whether it be the proposed method, super resolution microscopy, single molecule imaging, scattering microscopy, Raman microscopy or two photon imaging must serve a niche and specific purpose given that these different microscopy modalities have different capabilities, are used for imaging at different length and time scales, and have different sensitivities. What is the killer application that this method can be used for that none of the other methods would not be able to compete and what genuine biological insight this method could reveal that fluorescence (or even other label free) imaging cannot reveal.

In closing, I would like to thank the authors again for their patience as they read through my suggestions - the structural diversity of amyloids is a remarkable challenge in dementia research and I commend their interdisciplinary efforts in contributing to this challenge.

Reviewer #3 (Remarks to the Author):

The authors addressed all of my concerns and the paper is much more generalizable now. I recommend it for publication in Nat.Comm.

Label-free Identification of Protein Aggregates Using Deep Learning

Khalid A. Ibrahim, Kristin S. Größmayer, Nathan Riguet, Lely Feletti, Hilal A. Lashuel, and Aleksandra Radenovic

Manuscript #: NCOMMS-23-17657A

Response to Reviewers

Reviewer #2: I would like to thank the authors for their extensive response. I am convinced that the authors have made a commendable effort to open-source their codes and to clarify their rationale behind using the HTT cellular model in that study - and, therefore, happily recommend publication.

Reviewer #3: The authors addressed all of my concerns and the paper is much more generalizable now. I recommend it for publication in Nat.Comm.

We thank the reviewers for their positive assessment of our manuscript and for recommending publication.

We appreciate that Reviewer #2 is satisfied with the technical and experimental changes we have done for the revised manuscript and are happy to clarify the remaining minor points.

We have made every effort to thoroughly address any remaining comments. Please find enclosed a point-by-point response to each one.

The reviewer's comments are reproduced in black, **our answers are in blue, and changes to the manuscript are in green. *References to the manuscript text are in green italics.***

Reviewer #2:

1, the authors are not the first to develop a label-free method to detect large aggregates within a cell-free system, please see:

<https://www.ncbi.nlm.nih.gov/pmc/articles/PMC6239428/>

and

<https://www.nature.com/articles/s42003-021-01981-x>

and

<https://iopscience.iop.org/article/10.1088/2040-8986/ac5b51/meta>

First, we would like to point out that we did not make any claims that we are the first to develop a label-free method to image large aggregates. We had described that our method is the first to showcase the applicability of virtual labeling on unlabeled and unaltered protein aggregates in living cells.

We are aware of the possibility of studying protein aggregates using stimulated Raman scattering / Raman microscopy. In fact, we discussed previous work on label-free of protein aggregates and Raman microscopy in the introduction section of the manuscript, see below:

“Although recent studies have suggested using label-free methods, such as Raman microscopy^{32–35}, to monitor protein aggregation in cells, these methods lack the specificity and contrast needed to analyze them sufficiently, often lack the needed temporal resolution, or require deuterium labeling.”

We thank the reviewer for bringing these 3 additional papers on Raman and autofluorescence microscopy to our attention, **which are now cited in the revised version of the manuscript**. We have carefully reviewed the three papers highlighted by the reviewer and would like to point out the main differences in relation to our work.

We believe our method offers several major advantages over these approaches:

- 1) Our method has a much higher temporal resolution, particularly compared to Raman imaging. Ji et al. mention a frame rate of 5 s. Lochocki et al. and Ettema et al., mention a pixel dwell time of 177.32 μ s, corresponding to a rate of 1 frame per tens of seconds for a standard 50 μ m x 50 μ m cellular field of view.
- 2) LINA shows higher specificity and contrast. An example of the reduced specificity can be seen in Figure 2 of the work by Lochocki et al., where there is considerably more autofluorescence signal, compared to the ThS signal. Figure 3 and Figure 5 (e.g., #3c) also showcase this, where the authors describe the presence of a signal for Lipofuscin granular deposits. This reduced specificity is also shown in the other papers, for example in the bottom row in Figure 5 of the work by Ji et al., where there is signal present in the SRS image that is missing from the fluorescence image. Such signals can be misconstrued as aggregates if the fluorescence image is not there to be used as reference.
- 3) Related to point #2, the three papers do not quantify the percentage of false positives and negatives in their data, whereas we show very high accuracy of our models using different metrics and in various imaging conditions.
- 4) LINA provides the community with a simple method for automatic segmentation of protein aggregates, whereas none of the works reported in these papers offers a simple way to segment the aggregates that are being imaged. There is no data processing code provided.
- 5) Without the ability to automatically and accurately segment aggregates, quantitative analysis is impeded. For example, we show that measurements of aggregates' areas,

dry masses and intensity throughout formation are easily obtainable using our method. Similar quantitative analysis was not shown in the three studies.

6) None of the three papers show the possibility of using their label-free method on living samples, whereas we show the high applicability of LINA for live-cell imaging.

2, low affinity binders (and possibly a few high affinity binders such as antibodies) can target different conformations at the oligomeric, pre-fibrillar, and fibrillar levels of different proteins without altering (or substantially altering) the ultra-structure of the underlying aggregates allowing a very wide range of aggregates from small, nanoscopic oligomers to long, micro-scale sized fibrils to be imaged with high fidelity. A head to head comparison would show that such binders are easily capable of detecting a wider range of conformations across a longer length scale than the proposed label-free approach. It can be further argued that further use and development of the proposed approach will inevitably require the use of such binders to detect different conformations in a label-free manner.

We agree and have modified the text to include the following sentence:

Low affinity binders⁶¹ that could better capture the diversity of misfolded protein aggregates could have great potential to extend our method to enable the identification and label-free imaging of different aggregations states on the pathway to inclusion formation.

3, mentioning that the proposed method is insensitive to the conformational properties within the inclusions is an unsupported statement, first because the imaged conformations are a few (the strong conformational diversity of misfolded protein aggregates is a genuine bottle neck in this research field), and second, the neural network is not sufficiently interrogated to know what kind of features it has learnt to make it capable of distinguishing one type of aggregates from the other and whether those features are extendible.

We apologize for the lack of clarity. What we meant here is that while our method does not provide information about the conformations of mutant Httex1, it is not specific to a particular one and it is still able to detect mutant aggregates despite differences in their ultrastructure properties and proteome composition. For example, mutant Httex1-39Q inclusions have distinct ultrastructural properties compared to mutant Httex1-72Q inclusions and yet both are robustly detected by our method.

Indeed, the large diversity in the properties of misfolded protein aggregates is a complicated issue in this field. We have shown in our results that the model is generalizable to aggregates made from different constructs of Httex1, with varying morphologies, sizes, and masses. We aim in future work to address this issue in more detail by testing and developing even more conformation-insensitive models. Distinguishing between the different conformations (i.e., classifying them) is a separate task, which we believe machine learning would be highly suited for, but this beyond the scope of the current study.

4, this work could have been done through fluorescent tagging, as the authors indeed do. And, generally speaking, the use of different imaging modalities whether it be the proposed method, super resolution microscopy, single molecule imaging, scattering microscopy, Raman microscopy or two photon imaging must serve a niche and specific purpose given that these different microscopy modalities have different capabilities, are used for imaging at different length and time scales, and have different sensitivities. What is the killer application that this method can be used for that none of the other methods would not be able to compete and what genuine biological insight this method could reveal that fluorescence (or even other label free) imaging cannot reveal.

We thank the reviewer for giving us the opportunity to address this point.

At its simplest form, our models present a simple and rapid method for automatically finding and segmenting protein aggregates from transmitted-light images, which can be either quantitative phase images or the simplest-possible brightfield images. This provides users from labs around the world with an easy, high-throughput, automated technique to detect and segment aggregates, which has not been shown with any of the techniques mentioned by the reviewer. This can be done on live-cell images, enabling dynamic and kinetic studies, replicating the benefits of fluorescence microscopy, while avoiding the alterations and the phototoxicity it causes. This ensures that any information obtained is of high fidelity and is based on the true properties of the native protein aggregates. Furthermore, it is fairly simple to automatically measure the dry mass of different kinds of aggregates using our technique. Once images of aggregates are segmented, the benefits of any segmentation technique become attainable, such as counting (e.g., assessing the number of cells that have formed aggregates in different conditions, which can be used to test the effect of certain drugs), tracking and monitoring of aggregates, and quantitative analysis (area, morphology, intensity, spatial distribution, etc.).

Some of the biological insights include the possibilities to – with high specificity – study the colocalization of label-free aggregates with cellular organelles or other proteins, such as cytoskeletal proteins, without relying on confocal fluorescence microscopy of the aggregates themselves (and altering their properties). Another example is the ability to correlate our method with other label-free techniques, such as Brillouin microscopy, which currently rely on fluorescence microscopy to identify the aggregates, enabling fully label-free retrieval of the material properties of native aggregates.

Compared to Raman microscopy, our technique is much faster (we've shown that LINA works at a 3 ms exposure time, but it could theoretically go even faster). Another benefit is that LINA is much more specific, as we are able to directly detect aggregates in particular and not the soluble protein, as is the case for Raman microscopy. Our technique is very gentle, compared to Raman/scattering microscopy, super resolution microscopy, two photon imaging, and fluorescence imaging in general; of course, these techniques are very useful and still offer lots of advantages depending on the application.

To summarize, we believe that our technique has a lot of potential as a simple, fast, gentle, highly-specific, and automatic method for identifying, segmenting, and analyzing protein aggregates with high fidelity. This includes being able to measure the dry mass of aggregates and to monitor the kinetics of the aggregate formation process, without the need to fluorescently label the proteins, alter their properties and introduce toxicity.